# Intestinal stroma guides monocyte differentiation to macrophages through GM-CSF

Egle Kvedaraite [1,2,3] ✉, Magda Lourda [1,2], Natalia Mouratidou[4,5], Tim Düking [6], Avinash Padhi[1,7,8], Kirsten Moll[1], Paulo Czarnewski[9], Indranil Sinha [2], Ioanna Xagoraris[3,10], Efthymia Kokkinou[1], Anastasios Damdimopoulos[11], Whitney Weigel[1], Olga Hartwig [6], Telma E. Santos[6], Tea Soini[1], Aline Van Acker[1,12], Nelly Rahkonen[13], Malin Flodström Tullberg [1], Emma Ringqvist[1], Marcus Buggert[1], Carl Jorns [14,15], Ulrik Lindforss[16,17], Caroline Nordenvall[16,17], Christopher T. Stamper[1], David Unnersjö-Jess [18], Mira Akber[1], Ruta Nadisauskaite[1], Jessica Jansson[4], Niels Vandamme[19,20], Chiara Sorini [21], Marijke Elise Grundeken [8], Helena Rolandsdotter[22,23], George Rassidakis[3,10], Eduardo J. Villablanca [21], Maja Ideström[4,24], Stefan Eulitz[6], Henrik Arnell[4,24], Jenny Mjösberg [1], Jan-Inge Henter [2,25] & Mattias Svensson [1] ✉

Stromal cells support epithelial cell and immune cell homeostasis and play an important role in inflammatory bowel disease (IBD) pathogenesis. Here, we quantify the stromal response to inflammation in pediatric IBD and reveal subset-specific inflammatory responses across colon segments and intestinal layers. Using data from a murine dynamic gut injury model and human ex vivo transcriptomic, protein and spatial analyses, we report that PDGFRA⁺CD142⁻/low fibroblasts and monocytes/macrophages co-localize in the intestine. In primary human fibroblast-monocyte co-cultures, intestinal PDGFRA⁺CD142⁻/low fibroblasts foster monocyte transition to CCR2⁺CD206⁺ macrophages through granulocyte-macrophage colony-stimulating factor (GM-CSF). Monocyte-derived CCR2⁺CD206⁺ cells from co-cultures have a phenotype similar to intestinal CCR2⁺CD206⁺ macrophages from newly diagnosed pediatric IBD patients, with high levels of PD-L1 and low levels of GM-CSF receptor. The study describes subset-specific changes in stromal responses to inflammation and suggests that the intestinal stroma guides intestinal macrophage differentiation.

Stromal cells are mesoderm-derived non-epithelial cells that support integrity and functions in tissues of all organs. During recent years stroma and its subsets have emerged as key regulators of the immune response during homeostasis and tissue pathology, both in inflammatory and neoplastic contexts[1,2]. In intestinal mucosa, stromal cells are essential in maintaining the epithelial barrier and regulating immunity[3–5]. However, the role of the stromal compartment in shaping immune cells in the gut remains to be further defined.

Single-cell sequencing in patients with inflammatory bowel disease (IBD) has allowed the description of major stromal populations in the human intestinal mucosa, namely several subsets of fibroblasts, myofibroblasts, endothelium, pericytes, and glia[6–13]. Among platelet-

derived growth factor receptor A (PDGFRA)-positive fibroblasts, two major subsets (Stroma 1 and Stroma 2) were identified by Kinchen et al.[6], corresponding to WNT2B+ and WNT5B+ fibroblasts respectively, in the single-cell atlas from Smillie et al.[7]. WNT5B+ Stroma 2 expressed transcripts important for epithelial niche functions and, based on qualitative immunohistochemistry, was observed to be situated in proximity to the epithelium[6]. Data from adult[6–8] and pediatric[9–11] populations have laid the cornerstones of the transcriptional intestinal stromal landscape and its heterogeneity at single-cell resolution. However, our knowledge of spatial stromal cell distribution at the protein level across the different segments of the colon and across intestinal layers, as well as insights into their spatial co-localization with immune cells, remains limited.

With respect to inflammatory responses, a subset of fibroblasts associated with inflammation was described and referred to as Stroma 4, "inflammation-associated fibroblasts", and "activated fibroblasts" by Kinchen et al.[6], Smillie et al.[7], and Martin et al.[8], respectively. These activated cells showed transcriptional similarity to both WNT2B+ and WNT5B+ fibroblasts[7,11] and were suggested to reflect a distinct state, or program, along the crypt-villus axis[7]. In addition, it was suggested that activated intestinal stroma might predict non-responders to anti-TNF therapy in IBD[7,8,14], through synergy[14] or phenocopy[7] between oncostatin M (OSM) and TNF signaling leading to stroma activation. In the study by West et al.[14], the OSM receptor expression correlated positively with Podoplanin (PDPN), a glycoprotein associated with the Stroma 4 subset and shown to increase during inflammation in mice colitis and human adults with IBD[6,8,14]. Of note, Martin et al.[8] revealed additional receptor–ligand pairs, apart from OSM–OSM receptor, in myeloid cell–fibroblast crosstalk and found CD68+CD206− inflammatory macrophages in proximity to PDPN+ activated fibroblasts and suggested that inflammatory macrophages may activate fibroblasts. In the pediatric context, a small cluster with cells exhibiting transcriptional programs associated with inflammatory fibroblasts was detected in some, but not all, IBD patients[9,11]. It is, therefore, unclear if stroma activation is primarily associated with many years of disease and therapy-refractory IBD or is already detected at early stages of IBD.

Here, we quantify the inflammatory responses across stromal subsets and across different segments of the intestine in newly diagnosed pediatric IBD patients, as well as spatial stroma subset organization at the protein level across intestinal layers. In addition, cross-dataset validation in a dynamic gut injury model revealed a strong spatial association between fibroblast-monocyte/macrophages and transcriptomic/protein/spatial ex vivo evidence from the human intestine supported this observation. Finally, co-cultures of primary human fibroblasts and monocytes revealed that PDGFRA+CD142low/− intestinal fibroblasts guide monocyte differentiation to macrophages through GM-CSF.

## Results

### Intestinal stroma is activated at first diagnostic colonoscopy in pediatric IBD

To understand if stroma activation is detected at the early stages of IBD with minimal impact of confounding factors, the stroma activation status was investigated in newly diagnosed pediatric IBD patients (Fig. 1a and Methods section "Patients, samples, and disease scoring"). The overall PDPN expression in human colon was visualized in optically cleared biopsies that showed abundant expression of PDPN and the myofibroblast marker αSMA throughout the mucosa (Fig. 1b). The levels of PDPN were then quantified using conventional confocal microscopy, which demonstrated higher levels of PDPN in IBD patients compared to non-IBD controls (Fig. 1c). Flow cytometry on freshly isolated cell suspensions from mucosal biopsies revealed that, after excluding CD45+ immune cells and CD31+ endothelium, CD45−CD90+CD31− cells were either positive or negative for PDPN and expressed the pan-stromal marker Vimentin (VIM) (Fig. 1d). To assess

the PDPN levels specifically in colonic PDPN+ stroma and in relation to inflammation, matched biopsies from inflamed and non-inflamed areas of newly diagnosed pediatric IBD patients undergoing diagnostic colonoscopy were analyzed (Fig. 1a). The accuracy of the degree of inflammation in the matched inflamed and non-inflamed areas in colon was evaluated by comparing the macroscopic assessment performed by the pediatric gastroenterologist during the endoscopy procedure and the microscopic assessment obtained following histopathological examination (Fig. 1a). In addition, the level of tissue-infiltrating neutrophils, an established marker for tissue inflammation[15], was quantified in inflamed and non-inflamed areas using flow cytometry (Supplementary Fig. 1a) and was used to further confirm the degree of inflammation in the matched biopsies (Supplementary Fig. 1b and Methods). PDPN levels in PDPN+ stroma correlated with neutrophil counts in biopsies from patients with suspected IBD, which included IBD patients and non-IBD controls (Fig. 1e). Focusing on IBD patients, we observed higher levels of PDPN in inflamed mucosa, as defined based on endoscopic assessment (Supplementary Fig. 1c, left) and when neutrophil counts were considered (Fig. 1f and Supplementary Fig. 1d). Lastly, PDPN levels in PDPN+ colonic stroma correlated with mucosal disease severity (Supplementary Fig. 1c, right).

### Single-cell RNA-seq shows high levels of PDPN in all colonic fibroblast subsets during inflammation

To define the PDPN-positive stromal sources as well as overall stromal architecture upon inflammation, biopsies from macroscopically inflamed and non-inflamed mucosa from the same patient were sampled in treatment-naïve children newly diagnosed with IBD (Fig. 2a). Cells were freshly isolated from the biopsies taken during the first diagnostic colonoscopy, and using FACS sorting, stromal cells (CD90+) and epithelium (EPCAM+) were enriched and immediately subjected to a scRNA-seq pipeline (Fig. 2a and Methods). Analysis of the scRNA-seq data identified eleven cell clusters 0–10, corresponding to epithelium (clusters 1, 3, 5), four major fibroblast subsets (clusters 0, 2, 4, 6), myofibroblasts (cluster 7), glia (cluster 8), endothelium (cluster 9), and pericytes (cluster 10) (Fig. 2b and Supplementary Fig. 2a). RNA cluster distribution within each sample did not show any bias towards any specific individual (Supplementary Fig. 2b, c). As the proficient separation between the major stromal lineages is essential for robust downstream analysis, we further authenticated cell identities by assessing their global developmental relationships in the entire data set using Monocle[16], which confirmed the separation of fibroblasts and myofibroblasts from the remaining major clusters (Supplementary Fig. 2d). Cell identities were established based on key marker expression (Supplementary Fig. 2a) and differential gene expression profile[6] (Fig. 2c), and confirmed in an unsupervised manner using the Label Transfer function[17] with previously published data as a reference[6] (Supplementary Fig. 2e–g). Of note, cluster 6 was i) the smallest fibroblast cluster that, compared to other stromal clusters, included a relatively high number of mitochondrial genes and lower number of genes and molecules detected per cell, ii) was inconsistently labeled using the Label Transfer function; and thus compared to other stromal clusters, probably represented cells of poorer quality. We therefore decided not to validate/quantify this smaller fibroblast cluster at the protein level in downstream analyses.

Next, differentially expressed genes were identified between non-inflamed and inflamed samples in each cluster of cells individually (Supplementary Fig. 2h, i). This analysis demonstrated an inflammation-related global increase in expression of *PDPN* on all fibroblast clusters, corresponding to the previously described[6] stromal subsets 1, 2, 3/4, as well as myofibroblasts; but not on other stromal cells, such as glia, endothelium and pericytes Fig. 2d). In addition, PDPN+ fibroblasts (clusters 0, 2, 4, 6, 7) had a similar transcriptional response to inflammation as similar enrichment of pathways related to

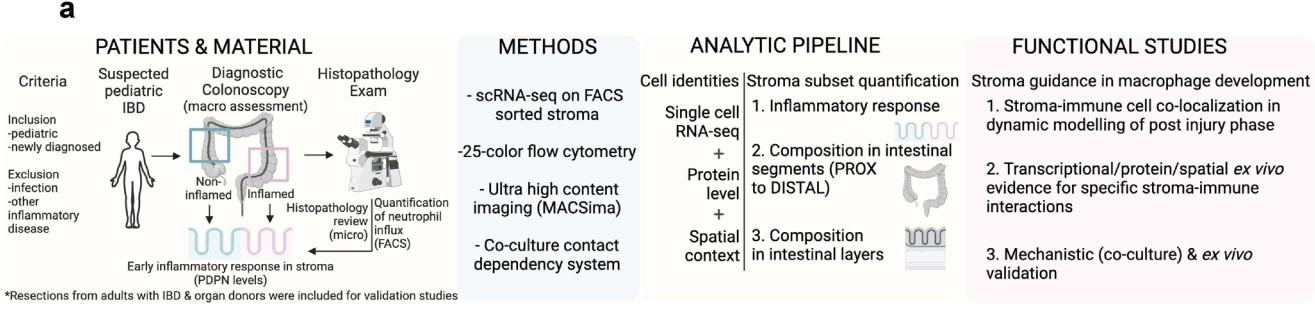

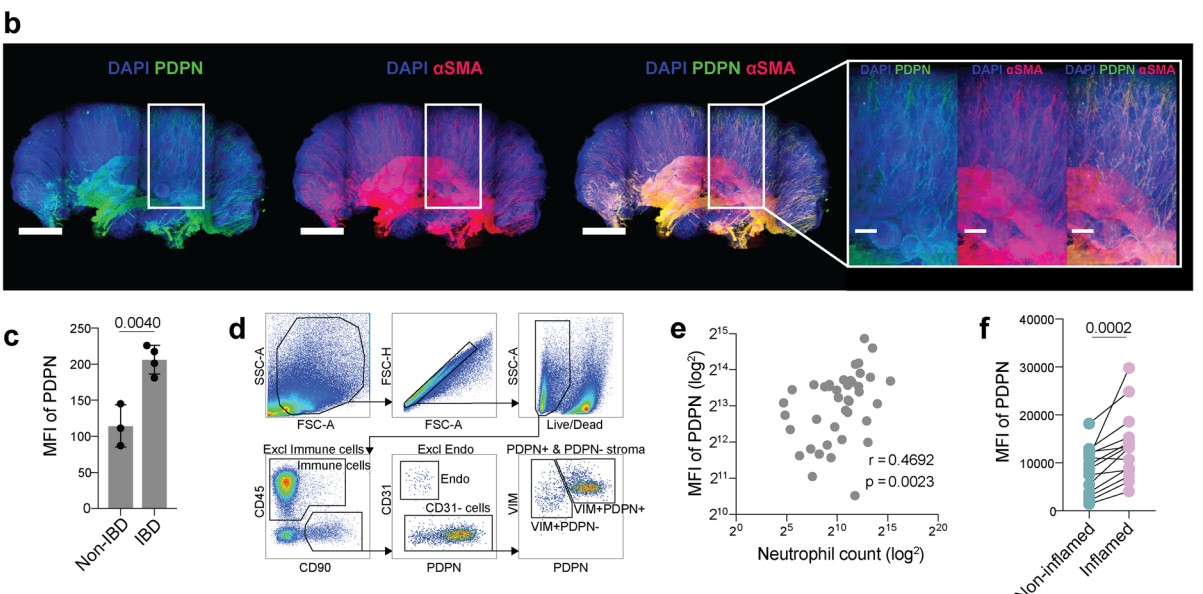

**Fig. 1 | Intestinal stroma is activated already at first diagnostic colonoscopy in pediatric IBD. a** Overview of overall study design. **b** Cleared mucosal biopsy from the healthy colon of a pediatric non-IBD patient (diagnosed with a colonic polyp), scale indicates 1 mm, higher magnification panel indicates 0.25 mm. **c** Levels of PDPN measured as mean fluorescence intensity (MFI) in colon biopsies from pediatric non-IBD patients ($n = 3$) and pediatric IBD patients ($n = 4$) using immunofluorescence microscopy. **d** Representative flow cytometry scheme for identification of PDPN+ and PDPN⁻ stroma in colon biopsies (newly diagnosed pediatric IBD, CD). **e** Correlation between PDPN levels in PDPN⁺ stroma and neutrophil counts in colon biopsies from children undergoing first diagnostic colonoscopy (see also Supplementary Fig. 1). **f** Levels of PDPN in PDPN⁺ stroma from non-inflamed and inflamed matched mucosal colon tissue in 14 newly diagnosed pediatric IBD patients; for inflamed/non-inflamed definition see also Supplementary Fig. 1. Mean with SD in (**c**), unpaired *t*-test (two-tailed *p*-value) for unmatched data (**c**) and Wilcoxon test for matched data (two-tailed *p*-value) (**f**) were used, and Spearman test (two-tailed p-value) was used to assess correlation (**e**).

fibrosis, chemokine, cytokine and JAK/Stat signaling was detected in these clusters (Fig. 2e).

## The composition of intestinal stroma varies across intestinal layers

To translate transcriptional heterogeneity to cellular and protein levels, the following criteria were applied for the selection of markers to be tested: (i) key markers that maintain differences in expression pattern in the absence or presence of inflammation; (ii) preferably extracellular markers that allow for sorting live cells; (iii) availability of well-performing commercially available antibodies. Based on this (Fig. 2f and Supplementary Fig. 3a) a 25-color flow cytometry panel (Supplementary Table 3) was designed allowing identification of CD56⁺CD9⁺ glia, CD31⁺PDPNʰⁱᵍʰ lymphatic endothelium, CD31⁺ endothelium, CD146ʰⁱᵍʰαSMAˡᵒʷCD9⁻ pericytes, αSMAʰⁱᵍʰCD130ˡᵒʷCD9⁺ myofibroblasts (MF), as well as PDGFRA⁺CD142⁺ fibroblasts (Fibro_4) and PDGFRA⁺CD142ˡᵒʷ/⁻ fibroblasts (Fibro_0_2) (Fig. 2g). To test the gating strategy taking the data from all used dimensions into account, cell gates were visualized in the UMAP space, that confirmed separation among the identified subsets (Fig. 2g, far right). Of note, the separation between Fibro_0 and Fibro_2 populations was attempted using HLA-DR but was not convincing due to the large overlap between the two in the UMAP space. Downstream analyses were therefore

performed on those subsets combined (referred to as Fibro_0_2). In addition, the gating was further validated by confirming marker expressions that were not used for gating the subset (Supplementary Fig. 3b, right). As expected, glia and endothelium expressed HLA-DR, Fibro_0_2 showed the highest levels of CD90 and CD130, while Fibro_4 expressed intermediate levels of αSMA and CD9 (Supplementary Fig. 3b). With respect to frequencies in the mucosa, all fibroblasts (Fibro_0_2 and Fibro_4), as well as myofibroblasts (MF) and glia, decreased during inflammation, while vascular endothelium (Endo), as well as lymphatic endothelium (Lymph. endo), increased (Fig. 2h). In a human setting with only small pieces of digested tissue it is challenging to do true count staining and get accurate cell counts for all the cell populations. Consequently, we report frequencies rather than absolute numbers, and the observed changes should be interpreted with caution, as they may reflect an expansion of a certain cell population, such as endothelium[18]. The overall composition across colon segments in matched samples was dependent on inflammatory status (non-inf vs. inflamed), with no striking differences detected across segments (cecum, ascendens, transversum, descendens, sigmoideum) (Fig. 2i). In contrast to the relatively stable composition in proximal to distal direction across colon segments, the frequencies of stromal cells differed across intestinal layers. Fibro_0_2 was detected at higher levels in submucosa and muscularis, compared to mucosa (Fig. 2j). In contrast,

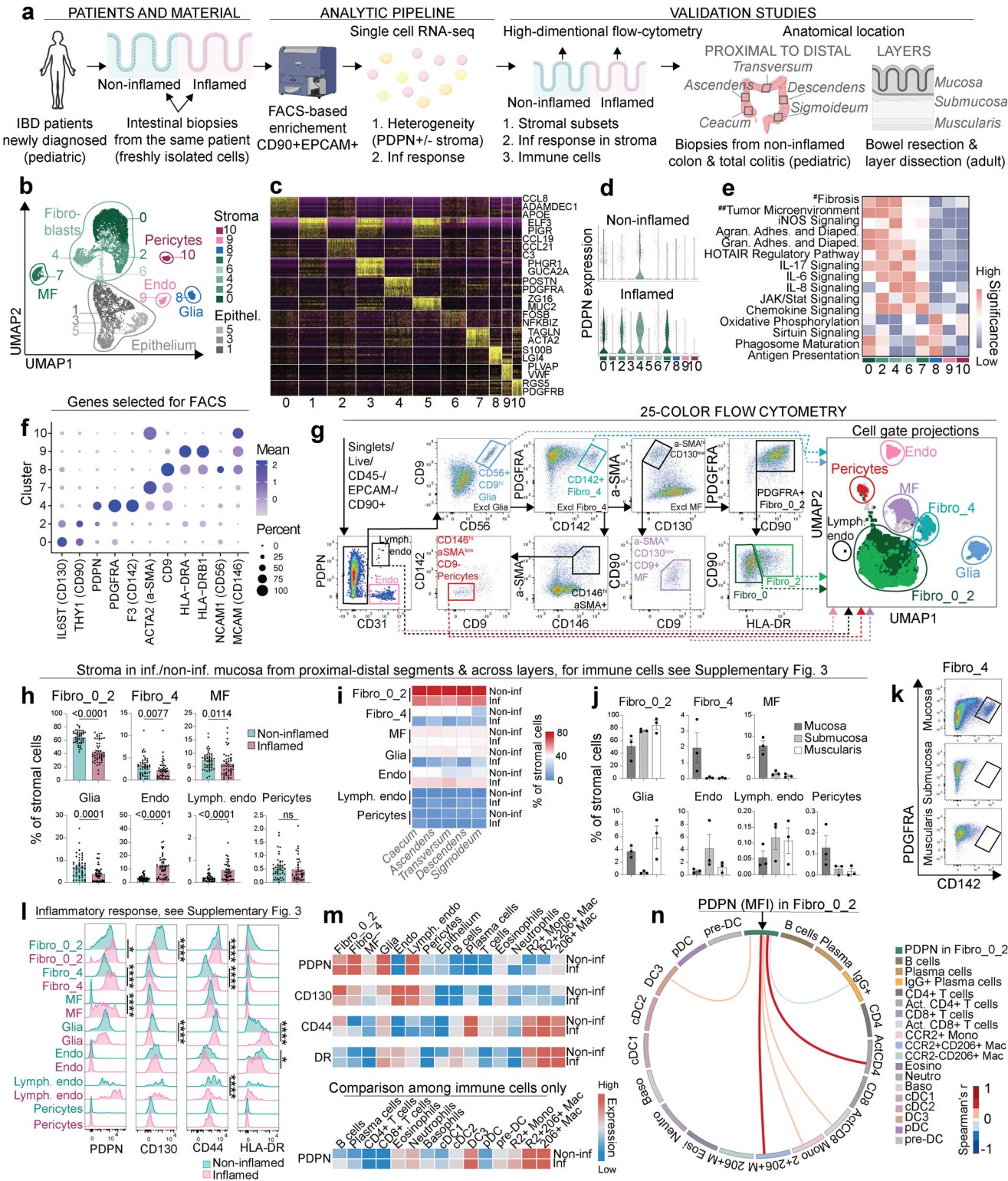

Fibro_4 was only detected in mucosa (Fig. 2j, k), where also MF was found at higher frequencies (Fig. 2j). In contrast, higher Fibro_0_2 frequencies were detected in submucosa and muscularis, compared to mucosa (Fig. 2j). As expected, endothelial cells were more frequent in submucosa (Fig. 2j).

These analyses show that stromal cell composition varies across intestinal layers in contrast to minimal variation across different colon segments investigated in the proximal to distal direction. Also, during inflammation in the mucosa, the frequency of fibroblasts and glia decreases while the frequency of endothelial cells increases.

**The inflammatory response in intestinal stroma is subset-dependent and associated with frequencies of specific immune cell populations**

Next, we aimed to quantify the inflammatory response across the stromal lineages on the protein level. For this, we selected four markers that have been implicated in the pathogenesis of IBD and other inflammatory conditions, namely PDPN, CD130 (IL-6 receptor), CD44, and HLA-DR[14,19–21]. These markers have previously been described to be associated with "inflammatory fibroblasts" or "activated fibroblasts"[1,6–8,22] and were measured across stromal subsets in

**Fig. 2 | Composition of intestinal stroma based on sc-RNAseq validated across colon segments and intestinal layers using 25-color flow cytometry. a** scRNA-seq design and validation analysis. **b** Clustering of scRNA-seq data showing clusters 0–10 in UMAP space. **c** Heatmap of top 50 differentially expressed genes among clusters 0–10. **d** Violin plots showing differential PDPN expression in clusters 0–10 from non-inflamed and inflamed samples. **e** Heatmap illustrating pathway significance, calculated as −log10(*p*-value) using right-tailed Fisher's Exact test, from IPA analysis on genes upregulated in inflamed compared to non-inflamed samples, assessed in each cluster independently; #Hepatic Fibrosis/Hepatic Stellate Cell Activation, ##Tumor Microenvironment Pathway. **f** Dot plot of gene expression, selected for validation on protein level, non-inflamed; for inflamed, see Supplementary Fig. 3a. **g** Gating strategy of stoma populations using FACS; gated populations shown in UMAP space see Supplementary Fig. 3b for more details. **h** Percentage of total stromal cells in 81–97 biologically independent non-inflamed and inflamed biopsies from pediatric patients undergoing first diagnostic colonoscopy. **i** Heatmap illustrating percentage of total stromal cells across the segments of the colon in matched samples from treatment naïve pediatric patients with no inflammation in the colon (Non-inf) or total colitis (Inf). **j** Percentage of total stromal cells across dissected intestinal layers from three adult IBD patients undergoing gut resection. **k** Representative flow cytometry plot of Fibro_4 subset across intestinal layers. **l** Levels of stromal activation markers across stromal subsets in non-inflamed and inflamed colon biopsies from children undergoing first diagnostic colonoscopy. **m** MFI levels of stroma activation markers in non-inflamed and inflamed colon biopsies from children undergoing first diagnostic colonoscopy, comparing levels across non-immune (epithelial, stromal) and immune cells (upper panels); and PDPN levels across immune cells only (lower panel), for gating strategy see Supplementary Fig. 3e. **n** Significant (*p* < 0.05) Spearman correlations between PDPN levels (MFI) in Fibro_0_2 and immune cell frequencies (two-tailed *p*-value). Mean with SE in (**h**) and (j); Mann–Whitney test was used to compare differences between two groups (two-tailed *p* value) (**h**, **l**), significance level for (**l**): ns > 0.05, *p < 0.05, **p < 0.01, ***p < 0.001 and ****p < 0.0001, for *p* values see Supplementary Fig. 3c.

inflamed and non-inflamed biopsies from newly diagnosed IBD patients (Fig. 2l). Flow cytometry data showed that even in non-inflamed mucosa Fibro_0_2 and Fibro_4 express PDPN (Fig. 2l), something that is not well-captured in the scRNA-seq data (Fig. 2d). Except for fibroblasts (Fibro_0_2 and Fibro_4), PDPN is also expressed on glia and lymphatic endothelium during steady state. However, while fibroblasts, as well as MF, which are negative during steady state, upregulate PDPN expression during inflammation, glia and lymphatic endothelium retain stable levels (Fig. 2l). Pericytes and vascular endothelium remain negative for PDPN during inflammation (Fig. 2l). Regarding the IL-6 receptor subunit CD130, it is downregulated during inflammation in Fibro_0_2 subset as well as in glial cells. A significant increase in CD44, which interacts with extracellular matrix components and induces activation of cell signaling pathways, was detected in Fibro_4, while in Fibro_0_2 as well as lymphatic endothelium, the CD44 levels were significantly decreased, even if the magnitude of the decrease was minor (Fig. 2l). In HLA-DR+ stromal cells, glia and endothelium, higher HLA-DR levels were detected in glia upon inflammation, while lower levels were observed in vascular endothelium (Fig. 2l).

Next, we compared how these markers, used to measure the inflammatory response across stromal subsets, were expressed and altered during inflammation in other cells within the tissue microenvironment (Fig. 2m). Compared with stomal cell subsets, PDPN was expressed on immune cells, such as granulocytes and macrophages, at low levels (Fig. 2m, upper panel). Focusing on immune cells only, the highest levels of PDPN were detected among CD206+ macrophages (both CCR2+ and CCR2low/−) and, unexpectedly, DC3 (Fig. 2m, lower panel). It has previously been suggested that infiltrating macrophages/monocytes activate fibroblasts[8], therefore, we investigated how the inflammatory response based on PDPN expression in the largest stromal subset Fibro_0_2, is related to frequencies of different immune cells (Fig. 2n), including activated T cell subsets (Supplementary Fig. 3e) and IgG+ plasma cells (Supplementary Fig. 3f), that are known to expand in IBD, and in our hands found at higher frequencies in inflamed tissue (Supplementary Fig. 3g). PDPN levels among Fibro_0_2 showed strongest correlations between CD206+CCR2+ macrophages and activated helper T cells (CD38+HLA-DR+CD4+ T cells). Other positive significant correlations included DC3, CD206−CCR2+ monocytes, and cytotoxic T cells (CD38+HLA-DR+CD8+ T cells) (Fig. 2n). Thus, inflammatory responses in the stroma are subset-dependent and in fibroblasts associated with expansion of specific immune cell subsets, such as CD206+CCR2+ macrophages and activated T cells.

## Ultra-high content imaging reveals the spatial composition of the intestinal stroma

To understand the stromal composition across intestinal layers, microanatomical landmarks, and inflammatory status, ultra-high content microscopy using the MACSima™ Imaging Platform (Supplementary Fig. 4a) was performed on inflamed resections from IBD patients and non-inflamed adult gut resection samples, that included resections from non-inflamed tissue from IBD patients and from organ donors, that were in both cases confirmed to be non-inflamed based on histopathology review (Fig. 3a and Supplementary Fig. 4b). After multi-parameter segmentation (Supplementary Fig. 5a), cell types were defined based on gating, and subsequent back gating to visualize the defined cell types in the images (Fig. 3b and Supplementary Fig. 5c). This allowed identification of Cytokeratin (Cyto)+ epithelium, immune cells: activated (CD38+) CD4+ and CD8+ T cells, CD20+ B cells, CD14+CD88+CD206−CD163− monocytes, CD14+CD88+CD206+CD163+ macrophages, CD66b+CD15+ granulocytes, CD138+CD38+ plasma cells, and stroma cells: PDGFRAB+PDGFRA+CD142low/− Fibro_0_2, PDGFRAB+CD142+ Fibro_4 situated close to the epithelium, CD31+Actin+ endothelium, αSMA+DES+ MF/muscle cells, and αSMA+CD146+ pericytes that were observed close to endothelium (Fig. 3b and Supplementary Fig. 5c). To further validate cell identity segregation, dimensionality reduction was performed, and projections of gated populations showed separation in the UMAP space (Fig. 3c). For the quantification of different cell types both frequencies and absolute numbers were calculated across intestinal layers and inflammatory status (Fig. 3d and Supplementary Fig. 5d). As expected, an overall increase in immune cells was seen in inflamed mucosa, with significant differences detected in monocyte, granulocyte, and B cell numbers (Fig. 3d). In submucosa, these differences were even more pronounced with elevated numbers of all immune cells examined, as well as, unexpectedly, Fibro_0_2 fibroblasts (Fig. 3d). In line with the flow cytometry data, Fibro_4 fibroblasts were only detected in mucosa (Fig. 3e). To quantify stromal cell distance to selected microanatomical landmarks, the distance to epithelium, endothelium or lymphatic endothelium was quantified (Fig. 3f). This confirmed that Fibro_4 were indeed located in close proximity to epithelium, and pericytes were found close to endothelium (Fig. 3f). This data provides a quantitative reference of spatial stroma cell distribution according to inflammatory status, as well as across intestinal layers and microanatomical location.

## Spatial stromal-immune relationship in dynamic modeling of gut healing reveals co-localization between fibroblasts and monocytes/macrophages

Next, we addressed intestinal stromal subset dynamics and their spatial and functional relationship with other cells in the microenvironment. Our data was aligned with human data covering all intestinal cell types from Smillie et al.[7] and spatial transcriptomics from a mouse gut healing model resembling changes seen in human IBD [23–25] (Fig. 4a). First, our stromal clusters were aligned to data from Smillie et al., confirming separation between endothelium, pericytes, glia, Fibro_4, and MF, while Fibro_0 and Fibro_2, in line with our observation on

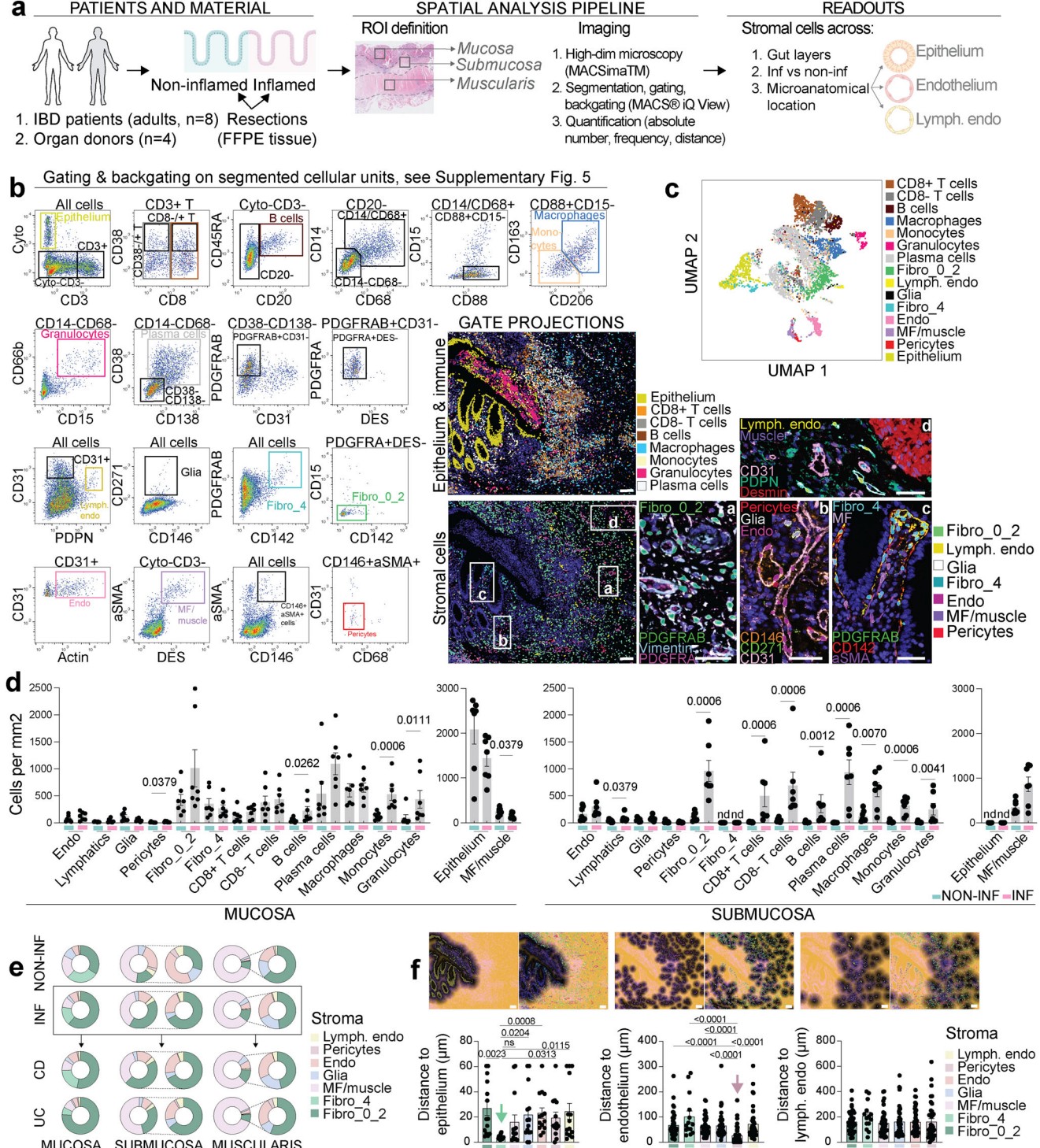

**Fig. 3 | Ultra-high content imaging reveals the spatial composition of intestinal stroma. a** Schematic overview of experimental design. **b** Gating strategy for identification of stroma, immune, and epithelial populations using ultra-high content MACSima™ Imaging Platform; gated populations are backgated to the image; scale bars indicate 100 μm in images on the left and 50 μm in images presented in higher magnification in areas (**a**–**d**); color codes for the lines defining gated cellular units are shown in upper left corner and color codes for marker signal are shown in lower left corner. **c** Gated populations projected in UMAP space. **d** Absolute numbers (cells/mm²) of cell subsets quantified across intestinal layers in 14 biologically independent non-inflamed and inflamed samples from adult gut resections; lymphoid tissue was excluded from the analyses. **e** Composition of stromal cells in non-inflamed and inflamed samples across intestinal layers. **f** Quantification of distance from stromal subsets to different microanatomical landmarks in 14 (epithelium) to 42 (endothelium, lymphatic endothelium) biologically independent samples; only mucosa was included in the calculation of the distance to epithelium; scale bar indicates 100 μm. Mean with SE in (**d**) and (**f**); Mann–Whitney test was used to compare differences between non-inflamed and inflamed samples (two-tailed *p* value) (**d**), and Kruskal–Wallis with Dunn's multiple comparisons test was used to comparing differences among stromal populations (**f**), ns not significant, nd not detected.

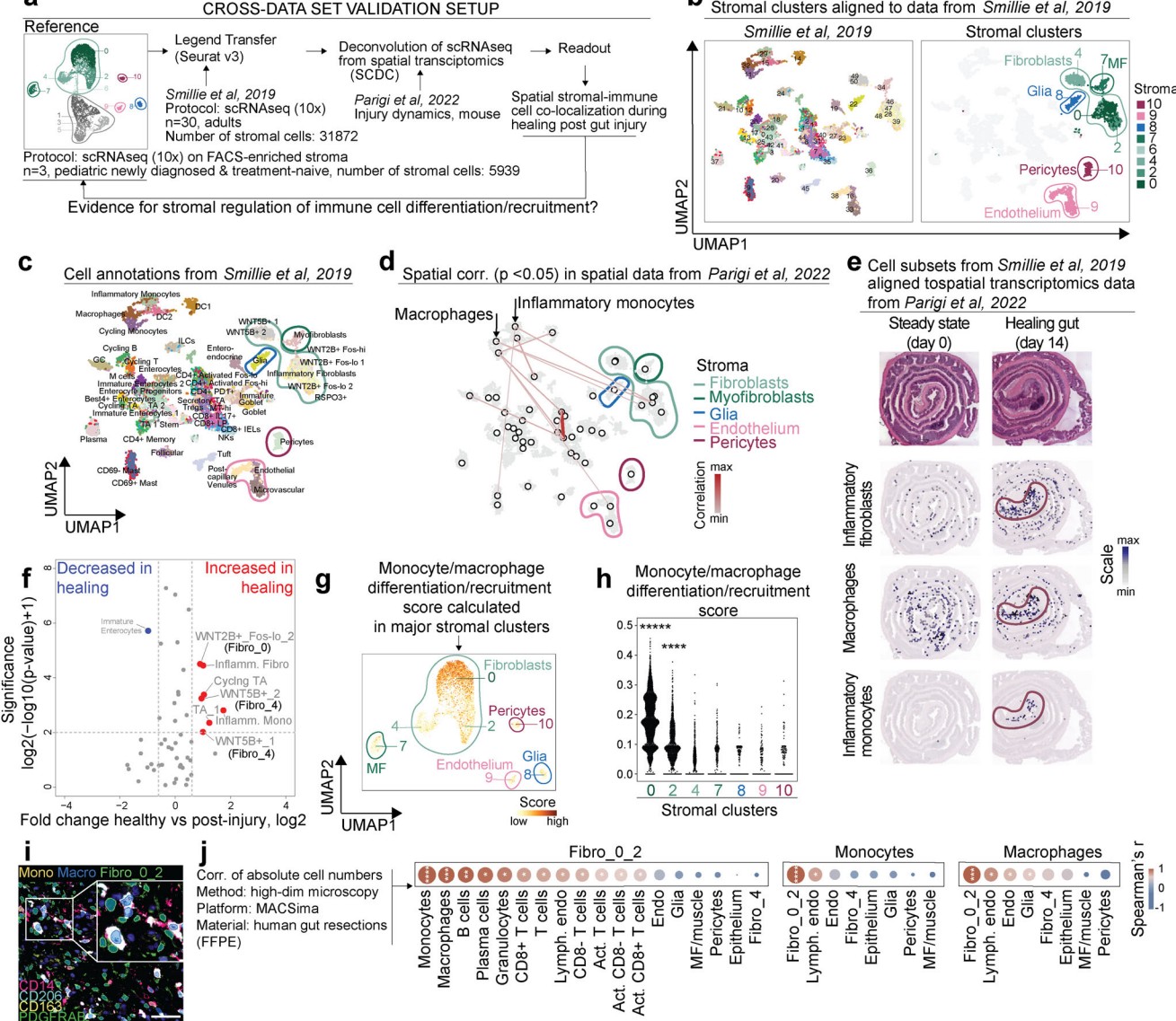

**Fig. 4 | Spatial stromal-immune relationship in dynamic modeling of gut healing reveals co-localization between fibroblasts and monocytes/macrophages. a** Schematic overview of cross-data set validation. **b** Stromal clusters aligned to scRNA-seq data from Smillie et al. using Label Transfer (see Online Methods). **c** Cell annotations in scRNA-seq data from Smillie et al. **d** Significant spatial correlations among cell subsets deconvoluted in spatial mouse colon data during the gut healing phase (day 14) from Parigi et al. **e** Inflammatory fibroblasts, macrophages, and inflammatory monocytes aligned to spatial transcriptomics data at steady state (day 0) or during the gut healing phase (day 14). **f** Volcano plot of human gut cellular composition deconvoluted from mouse steady state vs. post-injury phase (day 0 vs. day 14) from Parigi et al. **g** Monocyte/macrophage differentiation/recruitment score (CSF1, CSF2, IL33, IL34, IL6, CCL2, CCL7, CCL8, CCL13, IL3) calculated in stromal cell clusters, shown in UMAP. **h** Statistical

differences between clusters in the enrichment of Monocyte/macrophage differentiation/recruitment score. **i** Representative image from six independent experiments using ultra-high content imaging showing colors for staining in the lower left corner and segmented monocytes, macrophages, and Fibro_0_2 in the upper left corner; scale bar indicates 50 μm. **j** Correlations of absolute cell numbers from ultra-high content imaging experiments on human gut resections. Spearman $r$ was calculated for correlations (two-tailed $p$ value) (**j**), significance level: ****$p < 0.0000000005$, ***$p < 0.000005$, **$p < 0.00005$, *$p < 0.05$; Kruskal–Wallis test was used to compare differences between the clusters (**h**): ***** indicates adjusted $p$ value < 0.00001 for comparison to all other clusters, and **** indicates adjusted $p$ value < 0.00001 for comparison to all other clusters, except cluster 0.

protein level, showed partly overlapping pattern (Fig. 4b). As expected, annotations from Smillie et al. showed high concordance with our stromal cell identities (Fig. 4c). Spatial correlations between fibroblasts corresponding to Fibro_0 /Fibro_2 and macrophages/inflammatory monocytes were detected (Fig. 4d). The co-occurrence of deconvoluted cell identities of inflammatory fibroblasts, macrophages, and inflammatory monocytes was visualized onto Visium datasets (Fig. 4e), and changes were quantified, showing an increase of subsets corresponding to fibroblasts and inflammatory monocytes in the healing phase (Fig. 4f). Since this data suggests that inflammatory monocytes

and macrophages co-localize with Fibro_0 and Fibro_2, we next searched for transcriptomic ex vivo evidence of stromal regulation of monocytes/macrophages (Fig. 4a). To this end, we calculated an enrichment score encompassing relevant growth/differentiation factors (CSF1, CSF2, IL33, IL34, IL6, IL3) and CCR2 ligands (CCL2, CCL7, CCL8, CCL13). An enrichment was detected in Fibro_0 and Fibro_2 (Fig. 4g, h and Supplementary Fig. 6). We then compared the identified stromal regulation pattern on protein levels in human gut resections, where in contrast to scRNA-seq and/or flow cytometry-based approaches, quantification of absolute cell numbers can be

performed. Indeed, macrophages and monocytes were found in proximity to Fibro_0_2 (Fig. 4i), and the strongest correlations were detected between the absolute numbers of Fibro_0_2 and macrophages/monocytes (Fig. 4j). Together, this data shows a spatial fibroblast–monocyte/macrophage relationship in dynamic modeling of gut healing, supported by transcriptional ex vivo data from newly diagnosed and treatment naïve children with IBD (scRNA-seq, biopsies) and protein/spatial data from human gut resections (imaging analysis).

### Colonic PDGFRA⁺CD142^{low/−} fibroblasts guide monocyte to macrophage transition through GM-CSF

To investigate the effects of colonic fibroblasts on macrophage differentiation, a co-culture contact dependency system was established by culturing isolated healthy blood monocytes together with sorted PDGFRA⁺CD142^{low/−} colonic fibroblasts derived from the healthy colon (Supplementary Fig. 7), in a contact independent (transwell, TW), or contact-dependent manner, or alone (Fig. 5a and Methods). Concatenated flow-cytometry data from co-cultures were then subjected to unsupervised clustering using Phenograph, which was projected into UMAP (Fig. 5b). This revealed major differences in cluster distribution between the three culture conditions, with the largest differences between monocytes cultured alone and monocytes in-contact with fibroblasts (Fig. 5b–d). Clusters #9 and #1 corresponded to CD14⁺CD89⁺ monocytes, and #4 and #11 to CD14^{-/low}CD16⁺ non-classical monocytes. Clusters #14, #6, #8, #10, and #12 emerged in co-culture with primary fibroblasts and showed a tissue-resident macrophage phenotype, as evidenced by mannose receptor CD206 expression, including CD206⁺PDPN⁺ cells and PDL1^{+/−} clusters (Fig. 5e). CD206 induction was not restricted to macrophages developing in co-culture together with fibroblasts in a contact-dependent manner, but was also detected in transwell co-cultures (Fig. 5c, f). In addition to CD206, monocytes cultured in contact with fibroblasts had higher levels of the co-stimulatory molecule CD86 and the immune checkpoint molecule PDL1, and lower levels of the GM-CSF receptor (CD116), possibly indicating receptor-ligand ligation (Fig. 5f). As expected, the long-term tissue residency marker, FOLR2, associated with yolk-sac origin, was negative in all three culture conditions (Fig. 5f). Differences in monocytes/macrophages were governed by the presence of fibroblasts and to a lesser extent by the presence of the inflammatory stimuli TNF (Fig. 5g, h and Supplementary Fig. 8d, e). Since macrophage phenotype was acquired in a transwell system, even more clearly in the presence of TNF, the mechanism is not strictly contact-dependent, and we thus addressed the capacity of fibroblasts to produce soluble factors important for monocyte differentiation to macrophages. Indeed, fibroblasts produced IL-34, IL-33, GM-CSF, and M-CSF, the levels of which (except for M-CSF), increased in the presence of TNF (Fig. 5i). When blocking antibodies were used to neutralize these cytokines in the co-culture system, the tissue-resident phenotype with CD206 expression and frequencies of CD206 expressing clusters were diminished in the presence of anti-GM-CSF antibodies (Fig. 5j-l and Supplementary Fig. 8i). Sorted fibroblasts and monocytes/macrophages showed co-culture-mediated induction of *GM-CSF* mRNA in fibroblasts, but not in monocytes/macrophages (Supplementary Fig. 8f, g). Higher levels of soluble GM-CSF were confirmed in the contact-dependent co-culture (Supplementary Fig. 8h). In addition, co-culture induced gene expression of the CCR2 ligand *CCL2* in both cell types, along with higher mRNA levels of the proinflammatory cytokines *IL1b* and *IL6* in monocytes/macrophages (Supplementary Fig. 8f, g). Together this data shows that colonic PDGFRA⁺CD142^{low/−} fibroblasts can foster a tissue residency phenotype on monocytes through GM-CSF. Fibroblast effects on monocytes are not restricted to tissue residency (CD206, PDPN) or a tolerogenic phenotype (PDL1), as higher levels of pro-inflammatory cytokines are also produced by monocytes/macrophages in the co-culture.

### Ex vivo data support premises for colonic fibroblasts guidance of monocyte to macrophage transition through GM-CSF in the human colon

First, we confirmed the presence of GM-CSF in colonic PDGFRAB⁺ fibroblasts using double staining performed in human gut resection material and revealed higher levels of GM-CSF-producing PDGFRAB⁺ fibroblasts in IBD submucosa, compared to non-IBD submucosa (Fig. 5m and Supplementary Fig. 9). In addition, although detection of CSF2 was challenging on spatial transcriptional level, CSF2 transcripts were detected in semi-bulk sorted intestinal fibroblasts (Supplementary Fig. 10). The total monocyte/macrophage pool was then extracted from 25-color flow cytometry data on 96 samples from newly diagnosed pediatric IBD patients and subjected to dimensionality reduction using UMAP (Fig. 5n). CD206⁻ monocytes, CD206⁺FOLR2⁻ tissue-resident macrophages, and CD206⁺FOLR2⁺ long-term resident macrophages were identified by gating in UMAP (Fig. 5n, far left). Among the three subsets, a shift towards monocytes and a decrease of CD206⁺FOLR2⁺ macrophages were detected (Supplementary Fig. 8j). Indeed, the CD206⁺FOLR2⁻ macrophage population showed higher frequencies among immune cells in inflamed tissue in contrast to long-term tissue macrophages that declined (Supplementary Fig. 8k). Moreover, in line with the data from the co-culture system, lower levels of GM-CSF receptor (CD116) were detected in CD206⁺FOLR2⁻ tissue-resident macrophages, with lowest levels detected in long-term tissue-resident macrophages (Fig. 5o). PDL1 was expressed at higher levels in CD206⁺FOLR2⁻ tissue-resident macrophages compared to CD206⁻ monocytes (Fig. 5o), also confirming the co-culture data. CD86 expression was less conclusive, with significantly lower levels on long-term macrophages compared to both FOLR2⁻ subsets (Fig. 5p and Supplementary Fig. 8m). In line with co-culture findings, a PDPN increase coincided with higher degree of tissue residency (Fig. 5p and Supplementary Fig. 8m). No difference was observed in CCR2 expression between monocytes and CD206⁺FOLR2⁻ macrophages, that both exhibited significantly higher levels compared to long-term tissue resident macrophages (Fig. 5p and Supplementary Fig. 8m). Looking at possible sources of circulating monocyte subsets, the highest CCR2 levels were detected on classical monocytes, which also expressed highest levels of GM-CSF receptor (CD116) (Supplementary Fig. 8l). This shows that monocyte-derived cells fostered by fibroblasts through GM-CSF in co-culture resemble CD206⁺FOLR2⁻ tissue resident macrophages found in human intestine and implies circulating CCR2⁺GM-CSFR^{high} classical monocyte as a probable circulating source for CD206⁺FOLR2⁻ intestinal macrophages.

## Discussion

Stroma heterogeneity, composition, and spatial organization in homeostasis and inflammation, as well as its capacity to shape immune cells, are crucial for understanding chronic intestinal immunopathology. Here we revealed stromal subset-specific inflammatory responses in newly diagnosed pediatric IBD patients across different colon segments and spatial stroma subset distribution across intestinal layers. In addition, we found a strong spatial association between fibroblasts and monocyte/macrophages, and transcriptomic, protein and spatial ex vivo evidence from the human intestine supported this observation. Finally, using a co-culture system with primary human fibroblasts and monocytes we showed that PDGFRA⁺CD142^{low/−} intestinal fibroblasts guide monocyte differentiation to macrophages through GM-CSF.

The maintenance of cell population stability is a key element of tissue homeostasis, and the macrophage-fibroblast cell circuit has been described as stable and resilient to perturbations[26]. We provide

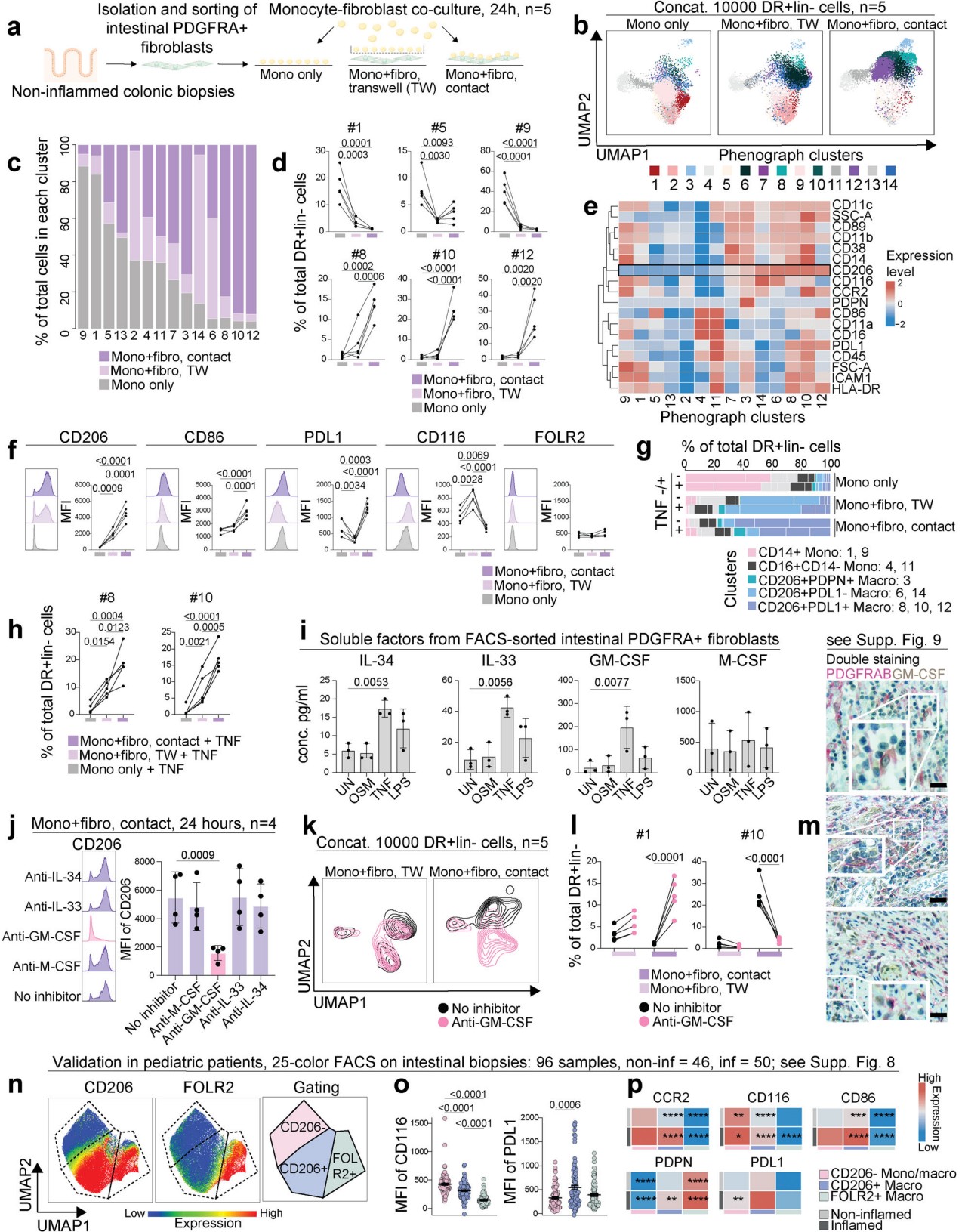

evidence for macrophage and specific fibroblast subset circuit stability in the gut based on: (i) strong spatial association in a dynamic gut injury model (spatial transcriptomics); (ii) strong correlation between absolute cell number of the two cell types (ultra-high content imaging, protein level); (iii) fibroblast enrichment in growth factors and chemokines important for monocyte/macrophage recruitment and differentiation (scRNA-seq data). By using a monocyte-intestinal fibroblast co-culture system, we provide functional evidence of PDGFRA⁺CD142^low/− fibroblast capacity to guide monocyte to macrophage transition through GM-CSF. Macrophage phenotypes derived from monocytes cultured in the presence of fibroblasts were diverse, with both pro-inflammatory and anti-inflammatory features. For

**Fig. 5 | PDGFRA⁺CD142^low/− fibroblasts guide monocyte to macrophage transition through GM-CSF. a** Overview of the co-culture, for gating see Supplementary Fig. 7. **b** Phenograph clusters of HLA-DR⁺lin⁻ cells in UMAP space across the three conditions. **c** Percentage of total HLA-DR⁺lin⁻ cells per cluster across the three conditions. **d** Cell frequencies across conditions in the three largest clusters of monocytes only condition (upper panels) or monocytes cultured with fibroblast conditions (lower panels). **e** Heatmap of marker expression across clusters. **f** Expression of selected markers across conditions in total CD11b⁺CD14⁺ monocyte/macrophage pool; same data ± TNF in Supplementary Fig. 8e. **g** Distribution and annotation of Phenograph clusters ± TNF. **h** Cell frequencies across conditions in selected clusters, +TNF. **i** Soluble factors in supernatants from sorted fibroblasts stimulated with OSM, TNF, and LPS for 24 h. **j** CD206 levels on monocytes/macrophages ± blocking antibodies against IL-34, IL-33, GM-CSF, or M-CSF. **k** Concatenated HLA-DR⁺lin⁻ cells in UMAP space; transwell (left) and contact (right) condition, ± anti-GM-CSF. **l** Cell frequencies across conditions of monocyte cluster

#1 and macrophage cluster #10, ±anti-GM-CSF. **m** Double staining of intestinal PDGFRAB⁺ fibroblasts and GM-CSF in inflamed submucosa of three representative patients with IBD, scale bar indicates 20 µm (see Supplementary Fig. 9). **n** Concatenated total intestinal CD11b⁺CD14⁺ monocytes/macrophages from 96 mucosal ex vivo biopsy samples in UMAP space; expression of CD206 and FOLR2, and gating strategy shown. **o** Quantification of GM-CSF receptor (CD116) and PDL1 in intestinal CD206⁻ monocytes/macrophages (pink), CD206⁺ macrophages (blue), and FOLR2⁺ macrophages (green) gated as shown in (**n**). **p** Heatmap of marker expression across intestinal monocyte/macrophage subsets in non-inflamed or inflamed samples. Data from three (**i**), four (**j**), five (**b, d, f, h–l**), and 96 (**o**) biologically independent samples were examined. Mean with SD in (**i**) and (**j**), mean with SE in (**o**); differences among three groups were assessed using one-way ANOVA with Holm–Šídák's multiple comparisons tests (**d, f, h, i, j, l**) and Kruskal-Wallis with Dunn's multiple comparisons tests (**o, p**); significance in (**p**): *$p < 0.05$, **$p < 0.01$, ***$p < 0.001$ and ****$p < 0.0001$; for $p$ values see Supplementary Fig. 8m.

example, higher levels of pro-inflammatory cytokines (e.g., *IL1B, IL6*) were detected in macrophages derived from co-cultures with fibroblasts, in line with transcriptomic evidence from recent IBD studies, implicating macrophage and fibroblast communication in drug unresponsiveness in adult patients with ulcerative colitis (UC)[7] and ileal Crohn's disease CD[8], where several receptor-ligand pairs mediating interactions between fibroblasts and macrophages were highlighted, including *IL1A, IL1B*, and *TNF* produced by macrophages[8]. On the other hand, we found higher levels of the immune checkpoint molecule PDL1 on macrophages fostered by fibroblasts in a contact-dependent manner. Data from intestinal biopsies in newly diagnosed children with IBD confirmed this phenotype with, among other markers, higher levels of PDL1 on intestinal CCR2⁺CD206⁺ macrophages compared to inflammatory monocytes. It remains to be understood how gut macrophage phenotype fostered by fibroblast-derived GM-CSF relates to their homeostatic functions, but it is interesting to note the promising results from trials where recombinant GM-CSF has been administered to IBD patients[27,28].

While our culture system cannot provide insights into the influx of CCR2⁺ monocytes, activated fibroblasts in IBD were previously reported to strongly express ligands for CCR2[8], a feature that we found to be specific for the PDGFRA⁺CD142^low/− fibroblast subset. It is, therefore, reasonable that CCR2⁺ classical monocytes, possibly recruited by PDGFRA⁺CD142^low/− fibroblasts, accumulate in the human intestine during IBD. Indeed, we detected a strong correlation between absolute PDGFRA⁺CD142^low/− fibroblast and monocyte/macrophage numbers. With respect to ontogeny, mouse data showed that the intestine-resident macrophage network is mainly supported and constantly replenished by circulating monocytes, rather than from embryonic sources, in a chemokine receptor CCR2-dependent manner[29,30]. In line with this, we detected re-distribution of the monocyte/macrophage pool with an increase in CCR2⁺CD206⁻ monocytes and CCR2⁺CD206⁺ macrophages and a decline in long-term tissue-resident macrophages positive for FOLR2, a marker possibly associated with the embryonic origin or time spent in tissue[31–33].

We show that PDGFRA⁺CD142^low/− fibroblasts guide monocytes in gaining a macrophage phenotype through GM-CSF, a growth factor with a well-established role in monocyte to phagocyte/macrophage differentiation[34–38], especially during inflammation[39,40], recently reviewed by Park et al[41]. Interestingly, it appeared that the presence of monocytes in contact with fibroblasts led to augmented GM-CSF production by PDGFRA⁺CD142^low/− fibroblasts, indicating a feedforward loop and monocyte-derived cells cultured in contact with fibroblasts showed a decrease in GM-CSF receptor, indicating active GM-CSF signaling. Indeed, in the presence of neutralizing anti-GM-CSF antibodies, the macrophage phenotype disappeared, and, importantly, ex vivo data from newly diagnosed IBD was in line with the co-culture result showing lower levels of GM-CSF receptor in intestinal CCR2⁺CD206⁺ macrophages compared to intestinal CCR2⁺CD206⁻

monocytes. While this data, together with GM-CSF (protein) positive fibroblast in the human intestine, describes a mechanism supported by in situ and ex vivo evidence in humans, in vivo insights would be valuable. For example, it would be interesting to further explore in vivo the role of GM-CSF producing intestinal fibroblasts found at higher levels in human IBD submucosa. However, developing the necessary conditional knockouts is a challenge, given the broad expression of stromal lineage markers. For example, targeting PDGFRA would hit more broadly than just one specific fibroblast cell population, as at least three functionally distinct PDGFRA+ stromal cell subsets have been described in mouse intestine[5].

Investigating the circulating monocytes[42], classical CD14⁺CD16⁻ monocytes had the highest levels of CCR2 and GM-CSF receptors. From a therapeutic point of view, the beneficial role of GM-CSF has previously been discussed[43], and a recent study showed that patients with neutralizing anti-GM-CSF autoantibodies had a higher risk of developing IBD[44], as well as a worse prognosis[45–48]. Our finding of elevated PDL1 on CCR2⁺CD206⁺ macrophages that developed in a GM-CSF-dependent manner contributes to an explanation of why neutralizing GM-CSF may lead to impairment of regulatory capacity in IBD. Indeed, the requirement of a sufficient PDL1 induction in orchestrating the immune response in the gut is also illustrated by the development of enterocolitis in cancer patients treated with PDL1 blockade, which often requires hospitalization and might be life-threatening[49]. In addition, anti-inflammatory treatments widely used in IBD show effectiveness also in checkpoint blockade-induced colitis[49]. On the other hand, the stromal capacity to shape macrophages might be of interest to explore in the pathologic immunosuppressive context as well since high levels of PDL1 have been described as a biomarker for a poor prognosis in colorectal cancer (especially in tumors with microsatellite instability)[50], and the primary source of PDL1 in colorectal cancer is tumor-associated CD14⁺ monocytes/macrophages[51]. Taken together, our data provide functional evidence for PDGFRA⁺CD142^low/− fibroblasts being potent modulators of macrophage development and warrant further investigation in a wider range of chronic inflammatory disorders.

In the era of single-cell technologies, spatial context and attention to detail while studying cell lineages and cell states are required. Although scRNA-seq approaches excel in revealing heterogeneity, the simultaneous detection of stromal cell lineages, their quantification, and their response to inflammation on the protein level provide the necessary translation to biological function. For example, different cell types expressing a strong common gene signature will cluster together, and examples of such signatures include proliferation, IFN-γ signature, Notch-dependent program across DC and monocyte lineages, and the newly described conserved DC program limiting anti-tumor responses[52–55]. Therefore, should the inflammatory fibroblast be considered a separate cell lineage or an activation program that may be acquired by multiple fibroblast lineages? Smillie et al. suggested the

latter, as in their data, both villus- and crypt-associated genes, corresponding to two major fibroblast subsets, were expressed by inflammatory fibroblasts[7]. We approached the fibroblast response to inflammation by quantifying markers associated with inflammatory/ activated fibroblasts across stromal subsets using 25-color flow cytometry, allowing separation of major stroma subsets. Interestingly, we found that inflammatory responses in different stroma cells are cell-type dependent. For example, both major fibroblast subsets, i.e., CD142[low/−] Fibro_0_2 and CD142[high] Fibro_4, as well as myofibroblasts, showed higher levels of PDPN in inflamed mucosa. One of the OSM receptor subunits, IL6ST (CD130), was detected at lower levels in inflamed mucosa, possibly indicating active signaling through this receptor, both on CD142[low/−] Fibro_0_2 and glia, but not on other stroma subsets. Interestingly, the strongest correlation between PDPN levels in CD142[low/−] Fibro_0_2 and immune cells was observed in CCR2[+]CD206[+] macrophages, further emphasizing the link between the two cell types, also with respect to the level of fibroblast activation. Overall, no major differences were detected in stroma composition across different colon segments, but instead across the intestinal layers. Frequencies of all fibroblasts and glia decreased in the mucosa, while endothelial cells, both vascular and lymphatic, were elevated as expected[18,56]. While CD142[low/−] Fibro_0_2 subset was detected across all layers and increased during inflammation in the submucosa, CD142[high] Fibro_4 was only detected in the mucosa, based on flow cytometry and high content imaging. The latter allowed distance quantification, which revealed that CD142[high] Fibro_4 had the shortest distance to epithelium compared to other stromal subsets. This data solidifies findings by Kinchen et al.[6] and Smillie et al.[7] on the Fibro_4 corresponding Stroma 2 and WNT5B+ fibroblasts, respectively, that transcriptionally resembled epithelium-supporting fibroblasts called telocytes[3,5]. While additional single-cell transcriptional and also epigenetic data from all the layers, including submucosa and muscularis, would indeed complement the overall picture of intestinal stromal cell heterogeneity, the ability to simultaneously discriminate between cell subsets at the protein level, e.g., using flow cytometry has biological relevance and is important for functional readouts in order to understand the pathogenesis of human diseases.

Homeostasis and inflammation represent a continuum of responses to tissue perturbations, and each tissue or organ is specialized to perform one or more essential functions to tackle such challenges[57]. Therefore, a comprehensive understanding of intestinal stromal cells, their heterogeneity, composition, and spatial organization, as well as their capacity to shape immune responses, is necessary to better understand and discover relevant treatment targets in IBD and other chronic inflammatory conditions. This study provides functional insights into the fibroblast capacity to shape monocyte transition to macrophages and generates a reference for intestinal stromal responses to inflammation, validating single-cell transcriptomic data on a protein level and in a spatial context.

## Methods

### Patients, samples, and disease scoring
Children undergoing their first diagnostic colonoscopy for suspected IBD were recruited at the Department of Pediatric Gastroenterology, Hepatology and Nutrition, Karolinska University Hospital, Stockholm, and Sachs' Children's and Youth Hospital, Södersjukhuset, Stockholm. Pinch biopsies were collected from these patients during routine colonoscopy and were matched with a blood sample for PBMC analysis, collected on the same day prior to colonoscopy. After clinical and pathologic evaluation, which included clinical exam, radiology, laboratory tests, endoscopy, and histopathology review, 48 patients were diagnosed with IBD, and patients that were not diagnosed with IBD were considered as controls ($n = 17$); patients with other inflammatory or infectious diseases were excluded. To control for mucosal inflammatory changes within the same patient, biopsies from a macroscopically inflamed area in the colon were matched with a macroscopically non-inflamed or less inflamed area, as defined during the colonoscopy (macroscopic assessment). Next, this definition was further evaluated using histopathological review, and areas were finally confirmed to be either inflamed or non-inflamed (microscopic assessment) (Supplementary Fig. 1). Clinical disease severity scoring was performed using PGA score[58], with the following categories: remission (0), mild (1), moderate (2) or severe (3) IBD. The mucosal disease severity was assessed using endoscopic scoring according to the Swedish Inflammatory Bowel Disease Register protocol[59], adapted from Mayo score[60], as follows for UC: score 0 (normal or inactive disease); score 1 (mild disease: erythema, decreased vascular pattern, mild friability); score 2 (moderate disease: marked erythema, absent vascular pattern, friability, erosions); score 3 (severe disease: spontaneous bleeding, ulceration); for CD: score 0 (normal or inactive disease); score 1 (mild disease: aphthous ulcers in otherwise normal mucosa); score 2 (moderate disease: swollen, erythematous mucosa, absent vascular pattern, friability, no ulcerations); score 3 (severe disease: edema, erythematous mucosa, absent vascular pattern, friability, ulcerations). To assess cellular composition throughout the colon (cecum−ascendens−transversum−descendens−sigmoideum) under steady state and inflammation, a unique clinical cohort was recruited to allow for matched comparisons: (i) children that after endoscopy turned out to have an entirely healthy colon, (ii) IBD patients with total colitis (meaning that the entire colon was equally inflamed). Details on clinical and laboratory characteristics are provided in Supplementary Tables 1 and 2. For analyses across intestinal layers, colon tissue was collected from adults with IBD ($n = 11$) undergoing resection surgery at the Karolinska University Hospital as well as organ donors ($n = 4$). The studies were approved by the Regional Review Board in Stockholm (2010/32-31/4, 2018/323-31/1, 2020-00507, 2019-05016), and written informed consents were obtained from patients and controls, as well as their parents for pediatric study participants.

### Tissue clearing and light sheet microscopy
Biopsies were fixed in 4% PFA for 2 h and were then washed in PBS for 1 h. The biopsies were then transferred to 3% agarose and sectioned using a PELCO easiSlicer (ProSciTech). Samples were then incubated at 50 °C in a clearing solution (200 mM boric acid, 4% SDS, pH 8.5) for 3 days. Next, samples were washed for 10 min in 1× PBS and transferred to primary antibodies diluted in PBST (1× PBS pH 7.5 + 0.1% Triton X-100) and incubated at 37 °C with shaking at 251 g for 5 days. Samples were then washed for 10 min in 1× PBS thrice and transferred to secondary antibodies diluted in PBST (1× PBS pH 7.5 + 0.1% Triton X-100) and incubated at 37 °C with shaking at 251 g for 5 days. Samples were then washed for 10 min thrice and transferred to Omnipaque (Apoteket) with a refractive index of 1.458 overnight before imaging. Samples were imaged using a Zeiss Z.1 light sheet microscope using a Clr Plan-Neofluar 20×/1.0 Corr nd = 1.45 (± 0.03) detection objective and 10×0.2NA illumination objectives). Details of antibodies used are provided in Supplementary Table 3.

### Immunofluorescence staining and confocal microscopy
Immunofluorescence staining was performed on frozen tissue as previously described[61]. Briefly, tissue sections were fixed in 4% PFA for 12 min, followed by blocking steps for unspecific staining using Image-iT FX Signal Enhancer (Life Technologies), Background Buster (Innovex Biosciences), and 10% FCS (Sigma-Aldrich, St. Louis, USA) in PBS with 0.1% saponin for 30 min. For intranuclear staining, the methanol permeabilization step was applied by incubating slides in ice-cold 100% methanol at −20 °C for 10 min, between the fixation and blocking steps described above. Next, the slides were incubated overnight at 4 °C with a mix of primary antibodies, followed by staining with a mixture of secondary antibodies and DAPI (Life Technologies) for

30 min and mounted in ProLong Gold Antifade Reagent (Life Technologies). Images were acquired using a Nikon A1R instrument and analyzed with the image analysis software Imaris (Bitplane, Zurich, Switzerland). Details of antibodies used are provided in Supplementary Table 3.

## Sample preparation for primary cultures, flow cytometry, and sorting

The protocol for splitting colon mucosa and submucosa was modified from that provided by Agace et al.[62]. Briefly, gut resection material from IBD patients was transported in wash buffer (HBSS $Ca^{2+}$ $Mg^{2+}$ gentamycin 10 mg/mL, pen-strep, fungizone 0.5ug/mL, HEPES 1 M) and processed within 3 hours of surgery. Mesenteric adipose tissue and the muscularis externa were removed by scissors. The colon tissue was then placed in a petri dish, mucosa side down, and the submucosa was peeled away using forceps with the aid of a stereomicroscope. Split tissue was examined for any contamination from either layer. Tissue pieces of 2–4 mm² were then cut into smaller pieces with scissors for digestion. DNase (0.25 m/mL) and collagenase (0.25 mg/mL) were added to IMDM + PS to digest tissue with a magnetic stir bar in a 37 C water bath for 45 min. The digestion was halted with the addition of 2× the total volume of IMDM + PS + 10% FCS.

To prepare cell suspensions from colonic biopsies, tissue biopsies were digested using 250 µg/ml DNase and collagenase II (Sigma) at 37 °C with magnetic stirring at 662 g for 25 min, followed by filtering through a 70-µm cell strainer. Bacterial and fungal contamination was prevented by first incubating cell suspensions in PBS supplemented with 500 µg/ml Normocin (InvivoGen), 0,5 mg/ml gentamycin (Thermo Fisher Scientific), and 2,5 µg/ml Amphotericin B (Thermo Fisher Scientific) for 10 min shaking. The cells were then washed once, and resuspended and plated in DMEM supplemented with 10% FCS, 2 mM L-glutamine (Thermo Fisher Scientific), 10 mM HEPES (Thermo Fisher Scientific), 1 mM sodium pyruvate (Thermo Fisher Scientific), 100 µg/ml Normocin (InvivoGen), 100 IU/ml penicillin (Thermo Fisher Scientific), and 100 µg/ml streptomycin (Thermo Fisher Scientific), further referred to as culture medium. After 2 h incubation at 37 °C under a 5% $CO_2$ atmosphere, non-adherent cells were washed away, and for the first passage, the culture medium was supplemented with 1 ng/ml PDGF-AB (Peprotech). Cells were then cultured for four to five passages until 80% confluence, and fibroblasts were sorted using the following gating strategy: CD45⁻EPCAM⁻CD90⁺CD31⁻PDGFRA⁺PDPN⁺/⁻ and collected in culture medium supplemented with 30% FCS instead of 10%. For stimulation, experiments sorted fibroblasts were cultured for 24 h or 72 h untreated or in the presence of recombinant human TNF 10 ng/ml, OSM 10 ng/ml (R&D Systems), or LPS 100 ng/ml or combinations of these stimuli at the indicated concentrations. PBMCs were isolated from blood samples after Ficoll separation using Lymphoprep (Stemcell Technologies). FACS staining was performed as previously described[63]. Briefly, cell suspensions were resuspended in PBS containing 2% FCS and 2 mM EDTA and a mixture of antibodies, supplemented with BD Horizon Brilliant Stain Buffer Plus (BD Biosciences) at 1:5 and FcR Blocking Reagent (Miltenyi Biotec) at 1:25 and were stained for 30 min. Next, cells were washed twice, fixed in PBS containing 2% PFA for 10 min, and washed again. For intracellular staining, after extracellular labeling and subsequent washing, cells were fixed for 30 min using Cytofix/Cytoperm (BD Biosciences) and stained with intracellular antibodies for 30 min, followed by two washes. Cells were acquired on a FACSymphony A5 or LSR Fortessa (BD Biosciences) and sorted using FACSAria Fusion Cell Sorter (BD Biosciences) or MA900 Multi-Application Cell Sorter (Sony Biotechnology). Details of antibodies used are provided in Supplementary Table 3.

## Flow cytometry analyses

Acquired data was analyzed using FlowJo v.10.5.3 (BD Biosciences), DIVA (BD Biosciences), and Prism v.8.0.2 (GraphPad Software Inc.). The algorithms used for dimensionality reduction were UMAP[64] (https://github.com/lmcinnes/umap) and Phenograph[65] (https://github.com/JinmiaoChenLab/Rphenograph). UMAP coordinates and Phenograph cluster annotation were assigned to each cell in each sample in the concatenated sample file and analyzed using FlowJo v.10.5.3 (BD Biosciences) and Prism v.8.0.2 (GraphPad Software Inc.). Marker expression was quantified in populations using geometric mean fluorescence intensity, and cell gates containing less than five cells were excluded from the analyses. Circos software package version 0.69-9 (http://circos.ca/) was used for the circular plot (Fig. 2n) using Spearman's $r$ correlation values between PDPN MFI in Fibro_0_2 and immune cell frequencies: % IgG⁺ of total plasma cells, % DR⁺CD38⁺ cells among CD4⁺ and CD8⁺ T cells for activated T cell subsets, % of MNPs for MNPs (Mono, Macro, DC), and % of total CD45⁺ cells for the remaining immune cell subsets.

## scRNA-seq

Cells were sorted into RPMI 1640 medium (Thermo Scientific, South Logan, US) supplemented with 10% FCS (Sigma-Aldrich, St. Louis, USA), 2 mM L-glutamine and 10 mM Hepes (Thermo Scientific, South Logan, US), using the BD FACSAria Fusion cell sorter, spun down and resuspended in PBS with 0.04% BSA at a final concentration of 1000 cells/µl. Cellular suspensions (target recovery of 10,000 cells) were loaded on the Chromium Controller (10× Genomics, Pleasanton) to generate single-cell Gel Bead-In-Emulsions. Libraries were prepared using Chromium Single Cell 3′ Reagent Kits v2 (10× Genomics) according to the manufacturer's instructions. Sequencing libraries were loaded at 2.1pM loading concentration on an HiSeq4000 with custom sequencing metrics (single-indexed sequencing run, 28/8/0/98 cycles for R1/i7/i5/R2) (Illumina, San Diego, Ca). Sequencing was performed at the VIB Nucleomics Core (VIB, Leuven, Belgium). Demultiplexing of the raw data and mapping to the human genome hg19 was done by the 10× CellRanger software (version 2.1.1; cellranger). Preprocessing of the data was done by the scran and scater R package according to the workflow proposed by Leucken and Theis[66]. Outlier cells were identified based on 3 metrics (library size, number of expressed genes, and mitochondrial proportion), and low-quality cells (low UMI counts, high % of mitochondrial genes) were removed from the analysis. Detecting highly variable genes, finding clusters, integrating, creating UMAP plots, and performing differential gene expression was done using the Seurat pipeline[17]. Label Transfer function in Seurat was used to confirm stromal cluster identity in an unsupervised manner, and Monocle 3 was used to define trajectories[16]. QIAGEN Ingenuity Pathway Analysis tool (also known as IPA) was used for pathway analysis. The null hypothesis for the analysis was that any molecules in our dataset do not overlap with a particular pathway. IPA uses right-tailed Fisher's Exact test to calculate $p$-values. A $p$-value less than or equal to 0.05 was considered significant. In this analysis, we first detected genes upregulated in inflamed compared to non-inflamed samples in each of the clusters and performed the IPA analysis for each. Next, we used a comparison analysis option in IPA to compare each of the IPA results from each cluster. A heatmap was generated by the IPA tool from where we have selected the most significant pathways, which were included in Fig. 2e. The enrichment score (Fig. 4g, h) of the gene set comprising of the genes: CSF1, CSF2, IL33, IL34, IL6, CCL2, CCL7, CCL8, CCL13, IL3 was calculated for each cell in R using the UCell package based on the counts of all genes. We also iterated through the gene set, calculating the enrichment score after removing one of the genes each time. To identify if there was any statistical difference in enrichment between the clusters, we used the multiple comparison test after Kruskal-Wallis from the pgirmess R package. The plotting was done in R using the ggplot2 package, and for the quasirandom plot, the ggbeeswarm package was used.

## Heat-induced epitope retrieval (HIER) and MACSima™ Imaging Cyclic Staining (MICS) technology using the MACSima

Prior to immunofluorescence staining, deparaffinization and rehydration were performed according to the manufacturer's protocol (Miltenyi Biotec B.V.& Co.KG). In brief, FFPE tissue sections were pre-heated for 1 h at 60 °C in a dry oven to soften the paraffin. Afterwards, tissue slices were placed in Xylol (# 9713.1; Carl Roth, Karlsruhe, Germany) for 20 min at RT. Deparaffinized tissue was rehydrated through a graded ethanol series (100%, 95%, 80%, 70%, 50%; # 100983; Merck, Darmstadt, Germany) and distilled water (1 min each).

HIER was performed in the Epredia Lab Vision PT-Module (# A80400012; Thermo Fisher, Waltham, USA) at 98 °C with TEC buffer (pH9) for 20 min. After cooling down (20 min at 85 °C), MACSwell™ Imaging frames were mounted directly on the slide and covered immediately with an appropriate volume of MACSima™ Running buffer (# 130-121-565; Miltenyi Biotec, Bergisch Gladbach, Germany). For indirect staining, tissue slices were first blocked using 5% (v/v) BSA (30 min at RT) followed by pre-incubation of unconjugated antibodies (CD206, PDGFRA) overnight at 4 °C. The next day, tissue slices were washed three times with MACSima™ Running buffer and subsequently stained with DAPI solution (a component of MACSima™ Stain Support Kit: # 130-127-574) in the dark for 10 min at RT. Finally, tissue slices were washed thrice with the MACSima™ Running buffer, and respective buffer was added to each well depending on the used MACSwell Imaging Frame.

As a fully automated instrument, the MACSima™ Imaging System (Miltenyi Biotec, Bergisch Gladbach, Germany) combines liquid handling with widefield microscopy for cyclic immunofluorescence staining. In brief, staining cycles consisted of the following automated steps: immunofluorescent staining, sample washing, multi-field imaging, and signal erasure by photobleaching. Incubation time was adapted to 30 min, and antibody concentrations were optimized individually. Unconjugated primary antibodies were detected using corresponding secondary antibodies during the normal process flow. Supplementary Table 4 summarizes the antibodies used for cyclic immunofluorescence staining with the MACSima™ Imaging Platform.

The acquired images by the MACSima™ Imaging Platform (single field of views) were preprocessed using MACSiQ® View Imaging Software (Analysis Module; Milteny Biotec) for stitching, registration and background subtraction according to the current Pre-Processing-Pipeline (Milteny Biotec). Preprocessed data was segmented to identify individual cells using DAPI (for cell nuclei) and CD3, CD142, CD14, PDGFRAB, CD68, CD138, Cytokeratin, CD31, CD15, Podoplanin, CD206 markers (for cell membrane) and analyzed using MACSiQ® View Imaging Software (Analysis Module; Milteny Biotec).

## Spatial deconvolution and analysis of single-cell clusters

To spatially map single cells to spatial transcriptomics spots, we made use of a previously described colonic single-cell dataset[7] to compile a representative reference for cell type deconvolution, where we randomly select 50 cells per cluster. This step is important to balance cell proportions and subtypes to allow a more accurate quantification using the weighted non-negative least squares regression (WNNLS) method implemented in the SCDC R package[67]. Two spatial transcriptomics datasets were used for deconvolution representing a colonic injury mouse model (GSE169749[23]). Mapping of cell types between the stromal dataset produced herein to the reference dataset[7] was done using KNN label transfer function implemented in Seurat package[17]. Additionally, we performed the detection of highly variable genes, principal component analysis, data integration with Harmony[68], and dimensionality reductions with UMAP[69,70] for visual projection of cell-cell spatial correlation results. Cell-cell spatial co-detection was estimated by Pearson correlations of the cell-type deconvolution results from SCDC, with only significant potential interactions being shown (FDR-corrected $p$-value < 0.05, and $r > \pm 0.1$). Finally, differential cell abundance comparison between the post-injury versus healthy tissue was calculated using the Wilcoxon test, where $p$-values below 0.05 and fold change above 1.5 were considered significant.

## Fibroblast-monocyte co-cultures in contact or in a transwell system

Colonic fibroblasts were grown until 80% confluency in the culture medium on TC-treated plates. Monocytes, isolated using negative selection with EasySept Human Monocyte Enrichment Kit without CD16 Depletion (Stemcell Technologies) from healthy donors, were added either directly on the fibroblast monolayer or on permeable inserts of 0.4 µm pore size Corning HTS Transwell system, $1 \times 10^4$ cells/well in 96 well and $5 \times 10^4$ cells/well in 24 well plate (Sigma). To mimic an inflammatory milieu, cells were cultured with or without 10 ng/ml recombinant human TNF (R&D Systems) for 24 h and were then harvested for flow cytometry analysis. To control for both contact-dependent and contact-independent effects, the same conditions were used to culture monocytes without fibroblasts present. For blocking experiments neutralizing antibodies against M-CSF, GM-CSF, IL-33, and IL-34 (R&D Systems) were added at 2 µg/mL.

## RNA isolation, cDNA preparation, and measurement of mRNA levels

The cells were sorted into 1.5 ml tubes with 350 µl of RNA lysis buffer containing 2-Mercaptoethanol. Total RNA from sorted cells was isolated using the Bioline isolate II RNA mini kit (Meridian Bioscience, Memphis, TN) as described in the manufacturer's protocol. The concentration and purity of RNA were measured using the spectrophotometer NanoDrop 1000 (Thermo Scientific, Waltham, USA). Based on the initial concentration, RNA was diluted with Nuclease-Free Water (Thermo Scientific) from 0 to 10 times. Total RNA (30 ng/sample) was used to generate cDNA using the SensiFAST cDNA synthesis kit from Bioline (Meridian Biosciences). The mRNA expression levels were measured with qPCR using Taqman Gene expression assays from Applied Biosystems, according to the manufacturer's protocol. The Taqman probe assays used were Hs00234140_m1 for CCL2, Hs00929873_m1 for GM-CSF, Hs01555410_m1 for IL1β, Hs00985639_m1 for IL6, Hs00174103_m1 for CXCL8 and Hs02786624_g1 for GAPDH. GAPDH was used as an internal control to normalize the results. Relative mRNA levels were calculated using the comparative threshold cycle (CT) method as described in Lourda et al.[71]. All samples were analyzed in duplicates, and the results are presented as relative mRNA expression using the formula $1 + 2^{-\Delta\Delta CT}$.

## Quantification of chemokines and cytokines

The cytokines and chemokines present in culture supernatants collected from stimulated or unstimulated monocytes and fibroblasts grown alone or in co-culture were measured using a human premixed 13-plex Luminex Discovery assay from R&D Systems, Minneapolis, USA. Briefly, samples were centrifuged at room temperature for 5 min at $16000 \times g$ to remove impurities and diluted (1:2) in calibrator diluent buffer. The unique standard cocktails were reconstituted and diluted 1:3 following the manufacturer's instructions. In total, 50 µL standard/sample was added and incubated for 2 h at room temperature. After that, microparticle cocktail beads were resuspended by vortexing, and 50 µL of the prepared suspension was added to each well and incubated for 2 h at room temperature with continuous shaking on a horizontal microplate shaker set at 1004 g. A microplate magnet was applied to the bottom, followed by triple washing with wash buffer. Next, 50 µL diluted biotin cocktail was added to each well and incubated for 1 h with shaking at 1004 g. Following a second washing step, 50 µL of diluted Streptavidin–PE was added, and incubated for 30 min at room temperature, followed by a third washing step. Finally, the microparticles were resuspended with 100 µL of wash buffer followed

by brief shaking of the plate for 2 min. The plate was read using a Luminex Magpix Microplate Reader (Luminex, CA, USA). For concentration calculations, the calibration curve for each molecule was analyzed with a five-parameter logistic curve.

## Double immunohistochemistry staining and analyses
Formalin-fixed, paraffin-embedded (FFPE) full intestinal tissue sections from patients with IBD were analyzed by immunohistochemistry (IHC). Antigen retrieval was achieved using heating with a cooking steamer for 40 minutes and the antigen retrieval reagent AR9 (cat. no. AR900250ML, Akoya Biosciences, Marlborough, MA, USA). A monoclonal rabbit anti-PDGFRAB antibody (cat. no. AB32570, Abcam, Cambridge, UK) at a dilution of 1:100, a monoclonal mouse anti-CSF-2 antibody (cat. no. TA808006, Waltham, MA, USA) at a dilution of 1:100 and the same concentration of IgG2b Mouse isotype control (cat. no. 557351, BD, New Jersey, USA) were used. Double immunostaining was performed using the ImmPRESS® Duet Double Staining Polymer Kit Peroxidase/Alkaline Phosphatase (HRP Anti-Mouse IgG/AP Anti-Rabbit IgG) (cat. no. MP-7724, Vector Laboratories, Burlingame, CA, USA) according to the protocol provided by the manufacturer. Hematoxylin was used as a counterstain. The LAS X Life Science Microscope Software Leica Application Suite X version 3.0.1423224 (Leica Microsystems) was used for analyses where pixels containing red, brown, or red and brown color were quantified. First, to confirm the specificity of the assay, the total area of the respective stain was quantified in $\mu m^2$ in IBD submucosa of matched subsequent sections comparing double staining with either CSF2 (brown) or Isotype control (brown). Next, a comparison was performed between the IBD patients and non-IBD donors (see Supplementary Fig. 9).

## Statistical analyses
Differences between two groups were evaluated using an unpaired *t*-test (Fig. 1c) for unmatched; and Wilcoxon test (Figs. 1f, 4f and Supplementary 1a, c, 9b) for matched groups. Mann–Whitney test was used to compare differences between two groups in Figs. 2h, l, 3d and Supplementary 3c, g, 5d, 8k, and 9d. Spearman test was used to assess correlations in Figs. 1e, 2n, 4j and Supplementary 1c; and the Pearson test was used in Fig. 4d. Differences among three groups were assessed using ANOVA with Holm–Šídák's multiple comparisons test in Fig. 5d, f, h, i, j, l and Supplementary 8a, c, e, h, i and Kruskal–Wallis test was used to compare differences among groups in Figs. 3f, 4g, h, 5o, p, Supplementary 5d, 8f, g, j, l, m. Analyses were performed in Prism v.8.0.2 (GraphPad Software Inc.). The significance for pathways analyses (Fig. 2e) was defined by the IPA software (Ingenuity Systems), as described in Methods.

## Reporting summary
Further information on research design is available in the Nature Portfolio Reporting Summary linked to this article.

# Data availability
All data associated with this study are available in the main text or the supplementary materials. scRNA-seq data have been deposited under accession number GSE169136, and bulk RNA-seq data under accession number GSE171582 in the Gene Expression Omnibus (GEO) database. Previously published data sets used in this paper are publicly available: GSE169749 for murine spatial transcriptomics and GSE158328 for human spatial transcriptomics. Source data are provided in this paper.

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

## Acknowledgements

We thank the patients and their families for their generous participation and all the medical staff at the Pediatric Gastroenterology Clinics for taking care of the patients, their recruitment, sample collection, as well as enthusiasm for translational research. We thank Steven Edwards and the Light Sheet Microscopy facility at KTH University and the National Microscopy Infrastructure, NMI (VR-RFI 2019-00217) for providing assistance in microscopy, as well as the core facility at NEO, BEA, Bioinformatics and Expression Analysis, which is supported by the board of research at the Karolinska Institutet and the research committee at the Karolinska University Hospital. Part of the data handling and analysis was enabled by resources at Project SNIC-2021-22-49, provided by the Swedish National Infrastructure for Computing, partially funded by the Swedish Research Council through Grant Agreement 2018-05973. PC was financially supported by the Knut Allice Wallenberg Foundation as part of the National Bioinformatics Infrastructure at Science for Life Laboratory Sweden. The study was supported by the Erik och Edith Fernströms Foundation, Sällskapet barnavård Foundation, Frimurare Barnhuset Foundation, The Swedish Childhood Cancer Fund, Dr. Åke Olsson Foundation for hematological research, The Region Stockholm (ALF-projects), The Knut and Alice Wallenberg Foundation, the Swedish Research Council (2023-03056), the Swedish Governmental Agency for Innovation Systems (VINNOVA) under the frame of NordForsk (Project no. 90456, PerAID), the Swedish Research Council under the frame of ERA PerMed (Project 2018-151, PerMIT), and Karolinska Institutet. Figures 1a, 2a, 3a, and 5a were created with BioRender.com.

## Author contributions

Conceptualization: E.K., M.S.; Investigation and/or formal analysis: E.K., M.L., N.M., T.D., A.P., K.M., P.C., I.S., I.X., E.f.K., A.D., W.W., O.H., T.S., Tea S., A.V.A., N.R., C.T.S., D.U.J., M.A., R.N., J.J., H.R., M.I., N.V., C.S., M.E.G.; Resources: M.F.T., E.R., M.B., C.J., U.L., C.N., G.R., E.J.V., J.M., S.E., H.A., J.I.H., M.S.; Writing—original draft: E.K.; Writing—review and editing: E.K., M.S.; Funding acquisition: E.K., M.L., J.M., J.I.H., M.S. All authors have provided feedback and approved the final version of the paper.

## Funding

## Competing interests

The authors declare no competing interests.

## Additional information

¹Center for Infectious Medicine, Department of Medicine Huddinge, Karolinska Institutet, Karolinska University Hospital, Stockholm, Sweden. ²Childhood Cancer Research Unit, Department of Women's and Children's Health, Karolinska Institutet, Stockholm, Sweden. ³Department of Pathology and Cancer Diagnostics, Karolinska University Hospital, Stockholm, Sweden. ⁴Pediatric Gastroenterology, Hepatology and Nutrition Unit, Astrid Lindgren Children's Hospital, Karolinska University Hospital, Stockholm, Sweden. ⁵Department of Medicine Solna, Karolinska Institutet, Stockholm, Sweden. ⁶Miltenyi Biotec B.V. & Co. KG, Bergisch Gladbach, Germany. ⁷Dermatology and Venereology Section, Department of Medicine Solna, Karolinska Institutet, Stockholm, Sweden. ⁸Division of Clinical Immunology, Department of Laboratory Medicine, Karolinska Institutet, Stockholm, Sweden. ⁹Science for Life Laboratory, Department of Biochemistry and Biophysics and National Bioinformatics Infrastructure Sweden, Stockholm University, Solna, Sweden. ¹⁰Department of Oncology-Pathology, Karolinska Institutet, Stockholm, Sweden. ¹¹Bioinformatics and Expression Analysis Core Facility, Department of Biosciences and Nutrition, Karolinska Institutet, Huddinge, Sweden. ¹²Tech Watch, Flanders Institute for Biotechnology, Ghent, Belgium. ¹³Integrated Cardio Metabolic Centre, Department of Medicine Huddinge, Karolinska Institutet, Huddinge, Sweden. ¹⁴Department of Transplantation Surgery, Karolinska University Hospital, Stockholm, Sweden. ¹⁵Department of Clinical Science, Intervention and Technology, Karolinska Institutet, Stockholm, Sweden. ¹⁶Department of Molecular Medicine and Surgery, Karolinska Institutet, Stockholm, Sweden. ¹⁷Department of Pelvic Cancer, GI Oncology and Colorectal Surgery Unit, Karolinska University Hospital, Stockholm, Sweden. ¹⁸Science for Life Laboratory, Dept. of Applied Physics, Royal Institute of Technology, Solna, Sweden. ¹⁹VIB Single Cell Core, VIB, Ghent, Belgium. ²⁰VIB-UGent Center for Inflammation Research, 9052 Ghent, Belgium. ²¹Immunology and Allergy Unit, Department of Medicine, Solna, Karolinska Institutet and University Hospital, Stockholm, Sweden. ²²Department of Clinical Science and Education, Södersjukhuset, Karolinska Institutet, Stockholm, Sweden. ²³Sachs' Children and Youth Hospital, Department of Gastroenterology, Södersjukhuset, Stockholm, Sweden. ²⁴Department of Women's and Children's Health, Karolinska Institutet, Stockholm, Sweden. ²⁵Theme of Children's Health, Karolinska University Hospital, Stockholm, Sweden. ✉e-mail: egle.kvedaraite@ki.se; mattias.svensson@ki.se

