## [Peer Review File · Nature Communications]

REVIEWER COMMENTS

Reviewer #1 (expert in gastroenterology):

This reviewer was supportive of publication prior to journal transfer.

Reviewer #2 (expert in innate immunity and inflammation):

This reviewer was not available for another review after journal transfer and was replaced by Reviewer #5.

Reviewer #3 (expert in stromal immunology and fibroblasts):

This reviewer was not available for another review after journal transfer and was replaced by Reviewer #4.

Reviewer #4 (expert in stromal immunology and fibroblasts):

The paper is well-suited to this journal.

My assessment of this work comes down to whether the in vitro fibroblast-driven mechanism in figure 5 is likely to accurately describe human biology. For me, figure 5M is the main data on this (double staining of fibroblasts and GM-CSF). The macrophage side of the analysis in figure 5 is helpful, but links between macrophages and GM-CSF will not alone be of broad enough interest. The question, for me, then is: Do gut fibroblasts express GM-CSF in vivo?

Figure 5M suggests they could, though I could not see this figure with enough resolution to really assess this. But it does not assess the expressing cell's phenotype (beyond a pan-fibroblast marker), or whether the provision of GM-CSF by this fibroblast subset is sufficiently frequent or unique to be convincing as a major or important source of GM-CSF, in a paper where other experimental systems are not provided.

In a paper centered on beautiful transcriptomics, readers are going to notice the lack of single-cell RNA-Seq or visium data showing fibroblast subset-level expression of GM-CSF.

I am assuming the human single-cell data does not provide a supportive addition, or the authors would have provided it. The authors should therefore look for another method (multiplexed histology staining?) to determine which fibroblast subsets express GM-CSF, and to assess frequency of expression. This will bring together the different parts of this paper and it will come closer to convincing readers about in vivo relevance of the main finding. In the absence of any targeted knockouts, organoid experiments, etc, is

expression of GM-CSF by that subset sufficiently frequent or unique to convince readers that this is likely to be relevant in vivo? I would not be prescriptive about how it's achieved.

I think the manuscript will not currently convince readers that a fibroblast source for GMCSF is likely to be important to macrophage differentiation in vivo. I believe readers will not be convinced by figure 5M

in its current form, and I am certain that all readers sufficiently skilled in the art will be wondering where the human single-cell RNA Seq data is to support the GM-CSF hypothesis in figure 5. For this reason, I think it would be wise to acknowledge the result, supportive or not.

Reviewer #5 (expert in gastroenterology, inflammatory bowel disease and mucosal immunology):

The real problem with this paper with 40 authors and a bucketful of data is that it is all a bit late. Alison Simmons paper on the same topic was in 2018, Smillie et al was in 2019, Huang et al was in 2019 and in 2022 there was Korsunsky et al and Jasso et al.

This reviewer doesn't really buy the notion that newly diagnosed IBD in children is somehow per se interesting because Huang et al also studied children. But this is also re-inforced by this reviewer's own experience in pediatric gastroenterology where I have seen many cases of 14 year olds coming in for their first colonoscopy whose colons are a mess with fistulae and even fibrosis.

But coming into this review process late, I looked very carefully at the initial reviews (which were excellent) and the authors responses and it seems to me that there have been quite revisions which have really improved the paper.

I don't know if the authors are aware of this but there have been trials of recombinant GM-CSF in IBD to try to boost macrophage function to destroy bacterial antigens in the tissues. This is based on the use of G-CSF in children with IBD associated with glycogen storage disease 1b

Point-by-point response to the reviewers' comments

Manuscript: Intestinal stroma guides monocyte differentiation to macrophages through GM-CSF

Reviewer #4 (expert in stromal immunology and fibroblasts):

The paper is well-suited to this journal.

Author response 1: Thank you for this comment.

My assessment of this work comes down to whether the in vitro fibroblast-driven mechanism in figure 5 is likely to accurately describe human biology. For me, figure 5M is the main data on this (double staining of fibroblasts and GM-CSF). The macrophage side of the analysis in figure 5 is helpful, but links between macrophages and GM-CSF will not alone be of broad enough interest.

Author response 2: Thank you for finding the macrophage analyses helpful. We would like to exemplify why our data is interesting to a broad audience. We not only show that GM-CSF induces monocyte differentiation to macrophages, but also show that colonic PDGFRA+CD142low/- fibroblasts guide this transition through GM-CSF. Fibroblast effects on monocytes are related to both tissue residency (CD206, PDPN), tolerogenic phenotype (PDL1), as well as their capacity to produce high levels of pro-inflammatory cytokines. This data provides important implications in understanding human IBD – e.g. patients with neutralizing GM-CSF antibodies have higher risk in developing Crohn's disease later in life (Mortha et al, Gastroenterology, 2022), and this might be related to inefficient fibroblast-driven GM-CSF-dependent generation of PD-L1+ macrophages. Indeed, neutralization of PDL1 in humans leads to colitis, similar to what we see in IBD (Ibraheim et al, Aliment Pharmacol Ther, 2020). Importantly, we validated this colonic macrophage phenotype (e.g. PDL1, PDPN, GM-CSFR) ex vivo in a cohort of newly diagnosed pediatric patients, specifically looking at monocyte-derived FOLR2- cells. These findings are interesting not only for IBD, but also for neoplastic contexts, where mechanistic insights into how PDL1 induction on gut macrophages is of key importance.

The question, for me, then is: Do gut fibroblasts express GM-CSF in vivo?

Author response 3: That's a valid point. Our ex vivo data strongly suggest that they do as we detect GM-CSF protein (Figure 5 and Extended Data Fig. 9). In addition, we could detect CSF2 mRNA when enough cells are analyzed in semi-bulk sorted gut fibroblasts (a couple of thousand cells sorted, and a relatively deep sequencing at an average 30 million uniquely mapped reads per sample) (Extended Data Fig. 10). Moreover, our findings are in line with previous literature showing that fibroblasts produce GM-CSF in different contexts (Stock et al, JEM, 2016; Chen et al, Hirota et al, Immunity, 2018; Eur J Immunol, 2018; Fuentelsaz-Romero, Front Immunol, 2021; Munker et al, Nature, 1986 and others summarized by Gasson, Blood, 1991), including the gut (Scheibe et al, Gut, 2017; Pang et al, Clin Exp Immunol, 1994).

Figure 5M suggests they could, though I could not see this figure with enough resolution to really assess this. But it does not assess the expressing cell's phenotype (beyond a pan-

fibroblast marker), or whether the provision of GM-CSF by this fibroblast subset is sufficiently frequent or unique to be convincing as a major or important source of GM-CSF, in a paper where other experimental systems are not provided.

Author response 4: High-resolution images including isotype control stainings, are provided (Extended Data Fig. 9). We do not claim that fibroblasts in general are, although they might be, the major source of GM-CSF in the gut. Instead we show: i) statistically significant correlation and co-localization between deconvoluted fibroblast and macrophage signatures in mouse visium data; and this was not detected with other cell types, including epithelium and T cells; ii) statistically significant enrichment of a score consisting of monocyte/macrophage chemotactic (e.g. *CCL2*) and growth factors (e.g. *CSF1*, *IL34*) in our Fibro_0_2 sc-RNAseq, which explains why those two cell types co-localize; iii) absolute numbers of those two cell types strongly correlate in high-content microscopy. We thus show that fibroblasts and monocytes/macrophages consistently share the necessary proximity and numbers for this mechanism to happen *in vivo*.

We test this in an experimental co-culture system where GM-CSF is both expressed and secreted by fibroblasts upon their interaction with monocytes, and neutralizing GM-CSF results in loss of macrophage phenotype in monocytes. Finally, we detect a similar phenotype in intestinal macrophages in newly diagnosed and treatment naïve children (*ex vivo*), including lower levels of GM-CSFR, just as the case for cultured macrophages.

In a paper centered on beautiful transcriptomics, readers are going to notice the lack of single-cell RNA-Seq or visium data showing fibroblast subset-level expression of GM-CSF.

Author response 5: It is not correct that GM-CSF single-cell RNA-seq data is lacking. This data has been included in the last two revisions (before and after transfer from Nature Immunology to Nature Communications), and it is still there. We apologize that this was not clear enough from the beginning, GM-CSF is coded by the gene “Colony Stimulating Factor 2” and thus the abbreviation for this gene is CSF2. Please have a look at Extended Data Fig. 6, named “*Individual gene expression from monocyte/macrophage differentiation/recruitment score (related to Fig. 4g, h)*”, included in line with previous requests from the Reviewer 4.

We have also searched for CSF2 transcripts in the published mouse and human visium data sets, but unfortunately it seems that CSF2 is just as hard to detect as in single-cell RNA-seq (Extended Data Fig. 10). However, when we went back and searched for CSF2 in semi-bulk sorted fibroblasts (a couple of thousand cells sorted, relatively deep sequencing at on average 30 million uniquely mapped reads per sample), we can indeed detect CSF2 also at the mRNA level (Extended Data Fig. 10).

I am assuming the human single-cell data does not provide a supportive addition, or the authors would have provided it. The authors should therefore look for another method (multiplexed histology staining?) to determine which fibroblast subsets express GM-CSF, and to assess frequency of expression. This will bring together the different parts of this paper and it will come closer to convincing readers about *in vivo* relevance of the main finding. In the absence of any targeted knockouts, organoid experiments, etc, is expression of GM-CSF by that subset sufficiently frequent or unique to convince readers that this is likely to be relevant *in vivo*? I would not be prescriptive about how it's achieved.

Author response 6: Please kindly see author responses 3, 4, and 5. Since scRNA-seq and visium based detection is challenging we performed double staining (protein) as well as analyzed bulk sorted fibroblasts (RNA). Based on flow cytometry, PDGFRA+ Fibro_0_2 fibroblasts account for the majority of stromal cells Fig. 2g, h, j, and PDGFRA is thus a relatively good marker for this subset. However, we agree *in vivo* exploration would be interesting and it has now been included in the discussion (page 19).

I think the manuscript will not currently convince readers that a fibroblast source for GMCSF is likely to be important to macrophage differentiation *in vivo*. I believe readers will not be convinced by figure 5M in its current form, and I am certain that all readers sufficiently skilled in the art will be wondering where the human single-cell RNA Seq data is to support the GM-CSF hypothesis in figure 5. For this reason, I think it would be wise to acknowledge the result, supportive or not.

Author response 6: Single cell data is present (Extended Data Fig. 6) and was there in the two last revisions. That not all cytokines are easily detected in sc-RNAseq is well known, and the visium data is also in line with this (Extended Data Fig. 10). Figure 5M has been further strengthened by additional images, isotype control staining, and quantification. We also hope the readers will consider the evidence presented in Figure 2n, Figure 4d, g, h, i, and j, as well as Extended Data Fig. 8, where co-localization *ex vivo*, correlation between absolute numbers *in situ*, as well as mechanism *in vitro* are described.

Reviewer #5 (expert in gastroenterology, inflammatory bowel disease and mucosal immunology):

The real problem with this paper with 40 authors and a bucketful of data is that it is all a bit late. Alison Simmons paper on the same topic was in 2018, Smillie et al was in 2019, Huang et al was in 2019 and in 2022 there was Korsunsky et al and Jasso et al.

Author response 6: Thank you for acknowledging the author list, this has indeed been a substantial effort over many years with a wide range of the necessary competences involved (human immunology, pediatric gastroenterology, GI surgery, single cell sequencing, high-dimensional flow cytometry, high-content imaging, primary co-culture systems etc). In contrast to studies mentioned by the reviewer, single cell sequencing represents just a little part of this work and should be considered as such. We acknowledge the atlas-like type of efforts in the studies mentioned above, and they are appropriately cited and discussed.

This reviewer doesn't really buy the notion that newly diagnosed IBD in children is somehow *per se* interesting because Huang et al also studied children. But this is also re-inforced by this reviewer's own experience in pediatric gastroenterology where I have seen many cases of 14 year olds coming in for their first colonoscopy whose colons are a mess with fistulae and even fibrosis.

Author response 6: Thank you for this comment. We agree there are reports where single-cell sequencing has been performed in pediatric cohorts, such as Huang et al, 2019, *Cell*; Elmentaite et al, *Nature*, 2021; Elmentaite et al, *Dev Cell*, 2020, and our recent study on ILCs

in pediatric IBD (Kokkinou et al, Cell Rep Med, 2023). None of them, however, enriched for stromal cells followed by validation on their data on protein level using multi-parameter (25 color) flow cytometry and high-content imaging analyses. These types of validations are of key importance to translate gene expression findings of stromal heterogeneity into the clinic, or address other stromal cell functions, e.g. their ability to shape monocyte differentiation to macrophages as we describe in our study.

For the second point, the cohort studied here is indeed unique and not reflective by the description “many cases of 14 year olds coming in for their first colonoscopy whose colons are a mess with fistulae and even fibrosis”. Having colonic fistulas is extremely rare in our centers, that are the largest center in the country (Malmborg et al, Inflammatory bowel diseases, 2015), meaning that patients with complicated disease are more likely to see our pediatric gastroenterologists. Nevertheless, none of the patients studied here had colonic strictures or fistulas at diagnosis. Other benefits with respect to biases in pediatric cohorts compared to adults are comorbidities and medication, that are common in adult cohorts.

But coming into this review process late, I looked very carefully at the initial reviews (which were excellent) and the authors responses and it seems to me that there have been quite revisions which have really improved the paper.

Author response 6: Thank you for the kind appraisal of our manuscript.

I don't know if the authors are aware of this but there have been trials of recombinant GM-CSF in IBD to try to boost macrophage function to destroy bacterial antigens in the tissues. This is based on the use of G-CSF in children with IBD associated with glycogen storage disease 1b

Author response 6: Thank you for bringing our attention to this, it has now been discussed (page 18).

REVIEWERS' COMMENTS

Reviewer #4 (Remarks to the Author):

The revisions have greatly strengthened this manuscript. While there is (always) more work that could be done, I believe this is a significant unit of work that is sufficiently rigorous for publication in this journal and will be valued by generalist audiences as well as specialists in gastroenterology and immunology.